# Exploration of Antarctic Ice Sheet 100-year contribution to sea level rise and associated model uncertainties using the ISSM framework

Nicole-Jeanne Schlegel[1], Helene Seroussi[1], Michael P. Schodlok[1], Eric Y. Larour[1], Carmen Boening[1], Daniel Limonadi[1], Michael M. Watkins[1], Mathieu Morlighem[2], and Michiel R. van den Broeke[3]

[1]Jet Propulsion Laboratory, California Institute of Technology, Pasadena, California, USA
[2]University of California, Irvine, Department of Earth System Science, Irvine, CA, USA
[3]Institute for Marine and Atmospheric research Utrecht (IMAU), Utrecht University, Utrecht, The Netherlands

*Correspondence to:* Nicole-Jeanne Schlegel
schlegel@jpl.nasa.gov

**Abstract.**

Estimating the future evolution of the Antarctic Ice Sheet (AIS) is critical for improving future sea level rise (SLR) projections. Numerical ice sheet models are invaluable tools for bounding Antarctic vulnerability; yet, few continental scale projections of century-scale AIS SLR contribution exist, and those that do vary by up to an order of magnitude. This is partly because model projections of future sea level are inherently uncertain and depend largely on the model's boundary conditions and climate forcing, which themselves are unknown due to the uncertainty in the projections of future anthropogenic emissions and subsequent climate response. Here, we aim to improve the understanding of how uncertainties in model forcing and boundary conditions affect ice sheet model simulations. With use of sampling techniques embedded within the Ice Sheet System Model (ISSM) framework, we assess how uncertainties in snow accumulation, ocean induced melting, ice viscosity, basal friction, bedrock elevation, and the presence of ice shelves, impact continental scale 100-year model simulations of AIS future sea level contribution. Overall, we find that AIS sea level contribution is strongly affected by grounding line retreat, which is driven by the magnitude of ice shelf basal melt rates and by variations in bedrock topography. In addition, we find that over 1.2 meters of AIS global mean sea level contribution over the next century is achievable, but not likely, as it is tenable only in response to unrealistically large melt rates and continental ice shelf collapse. Regionally, we find that under our most extreme 100-year warming experiment generalized for the entire ice sheet, the Amundsen Sea Sector is the most significant source of model uncertainty (1032 mm $6\sigma$ spread), and the region with the largest potential for future sea level contribution (297 mm). In contrast, under a more plausible forcing informed regionally by literature and model sensitivity studies, the Ronne basin has a greater potential for local increases in ice shelf basal melt rates. As a result, under this more likely realization, where warm waters reach the continental shelf under the Ronne ice shelf, it is the Ronne basin, particularly the Evans and Rutford Ice Streams, that are the greatest contributors to potential SLR (161 mm) and to simulation uncertainty (420 mm $6\sigma$ spread).

# 1 Introduction

The Antarctic Ice Sheet (AIS) contains the vast majority of Earth's surface water mass, outside of the oceans ($\sim 65$ m of sea level rise equivalent). Over the past three decades, observations of the AIS have uncovered changes in ice dynamics responsible for significant increases in regional mass loss (Velicogna, 2009; Rignot et al., 2011; Pritchard et al., 2012; Shepherd et al., 2012; Gardner et al., 2018). Estimation of the ice sheet's future evolution over the coming centuries and its consequential contribution to sea level change have therefore become subjects of many studies, as they are critical to the improvement of sea level projections. Model based studies that rely on the numerical representation of physical processes driving changes in ice sheet mass are the best option to simulate the future evolution of the AIS. However, results from such studies significantly deviate in magnitude, ranging from up to 30 cm of global mean sea level equivalent (SLE) contribution by the end of the $21^{st}$ century (e.g., Ritz et al., 2015) for a midrange emissions scenario, to over a meter of SLE (e.g., DeConto and Pollard, 2016) for a high-end warming scenario. Model-based estimates of ice sheet mass balance are associated with large uncertainties that are difficult to quantify. Even the most state-of-the-art ice sheet models have limitations related to poorly characterized or resolved physical processes, and they also include large uncertainties in external forcing, low resolution continental scale simulations, poorly known initial and boundary conditions, or uncertainties in external forcing. All of these contribute to the large spread amongst current ice sheet projections (e.g., Nowicki et al., 2013).

Community efforts like the Sea-level Response to Ice Sheet Evolution (SeaRISE) initiative have adopted multi-model ensemble approaches in order to leverage the strengths and limitations of several ice flow models (Bindschadler et al., 2013; Nowicki et al., 2013) and to quantify uncertainties associated with ice sheet model projections. However, the participating models relied on different initialization procedures, included varying physical processes, and also applied external forcing in different ways, such that the sources of model uncertainties could not be confidently distinguished. Assessing the uncertainty caused by input errors is a major challenge being faced by the ice sheet model community, and previous modeling-based studies mainly relied on ensemble simulations (e.g. Winkelmann et al., 2012; Ritz et al., 2015; DeConto and Pollard, 2016; Bakker et al., 2017) to estimate the impact of some unknown model parameters and forcing. Recent developments and improvements in ice flow model efficiency now allow the modeling community to utilize uncertainty quantification (UQ) to explore how errors in various model boundary conditions and forcing propagate in ice flow models (Larour et al., 2012b, a; Schlegel et al., 2013, 2015).

In this study, we use the UQ tools from the DAKOTA (Design Analysis Kit for Optimization and Terascale Applications) framework (Eldred et al., 2008), embedded within the Ice Sheet System Model (ISSM), to study the impact of uncertainties in surface mass balance, ocean induced melting, ice viscosity, basal friction, bedrock geometry, and the presence of ice shelves, on global mean sea level (GMSL) projections over the next century. Understanding the impact of the initial conditions is being extensively investigated as part of the Ice Sheet Model Intercomparison Project for CMIP6 (ISMIP6, Nowicki et al., 2016; Goelzer et al., 2018) and therefore is not be considered in this study; instead we focus on the other sources of model uncertainty.

We first describe the Antarctic ice flow model, the methodology applied, as well as the datasets used. We then detail the experiments performed and present the results of the UQ sampling simulations. Finally, we discuss the results and their implications for sea level projections over the next century.

## 2  Methods and Data

### 2.1  Model description and initialization

ISSM is a thermo-mechanical finite-element ice flow model. It relies upon the conservation laws of momentum, mass, and energy, combined with constitutive material laws and boundary conditions. The implementation of these laws and treatment of model boundary conditions are described by Larour et al. (2012c). For the experiments described here, the choice of ice flow stress balance approximation is described below. The rheology law is based on Glen's flow law (Glen, 1955) (where Glen's flow law exponent has a value of 3), with the ice viscosity depending on the ice temperature (Cuffey and Paterson, 2010), after Larour et al. (2012c) and Seroussi et al. (2017b). The basal friction is based on a Budd-type friction law (Budd et al., 1984) such that the basal velocity vector tangential to the glacier base plane ($\mathbf{v_b}$) and the tangential component of the external force ($\boldsymbol{\sigma} \cdot \mathbf{n}$) follow the relation $\boldsymbol{\tau_b} = -\alpha^2 N \mathbf{v_b}$ where $\alpha$ is defined as our basal drag coefficient (Cuffey and Paterson, 2010). Here, in the absence of a hydrological model, assuming a perfect connection with the ocean and no hydrological flow, we approximate $N$, the effective pressure of water at the glacier base, as $N = g(\rho_i H + \rho_w z_b)$ (Cuffey and Paterson, 2010), where $g$ is gravity, $H$ is ice thickness, $\rho_i$ is the density of ice, $\rho_w$ is the density of water, and $z_b$ is bedrock elevation relative to sea level. Note that at sea level, $z_b$ is zero and that below sea level, it is negative. The grounding line evolves assuming hydrostatic equilibrium and following a sub-element grid scheme (SEP2 in Seroussi et al., 2014b). The ice front remains fixed in time during all simulations performed, and we impose a minimum ice thickness of 1 m everywhere in the domain.

The model domain covers the entire AIS as observed today, and its geometry is interpolated from the Bedmap2 dataset (Fretwell et al., 2013), with specific exception to the Amundsen Sea sector, Recovery Ice Stream, and Totten Glacier. In these areas, bed topography is mapped at 150 m spatial resolution using the mass conservation (MC) technique described by Morlighem et al. (2011) and published in Rignot et al. (2014) for the Amundsen Sea Sector (hereafter referred to as Bedmap2/MC). The simulations rely on a mesh resolution varying between 1 km at the domain boundary and within the shear margins, and 50 km in the interior, with a resolution of 8 km or finer within the boundary of all initial ice shelves, leading to a 187,445 element anisotropic triangular mesh (Fig. S1).

In order to estimate land ice viscosity, we compute the ice temperature based on a thermal steady state with 15 vertical layers (Seroussi et al., 2013). This thermal simulation uses three dimensional higher-order (Blatter, 1995; Pattyn, 2003) stress balance equations, observations of surface velocities (Rignot et al., 2011), and basal friction inferred from surface elevations (Morlighem et al., 2010). The boundary conditions of the thermal model are geothermal heat flux and surface temperatures from Maule et al. (2005) and Lenaerts et al. (2012) respectively. The resulting ice temperatures are vertically averaged, used as inputs in the ice flow law, and held constant over time, after Schlegel et al. (2015). While this assumption of a steady state

thermal regime is not exact (as the ice sheet is not in thermal equilibrium), it has been proven to have a limited impact on century-scale ice flow simulations (Seroussi et al., 2013).

To infer the unknown basal friction coefficient over grounded ice and the ice viscosity of the floating ice, we use data assimilation (MacAyeal, 1993; Morlighem et al., 2010), to reproduce observed surface velocities from Rignot et al. (2011). Then, we run the model forward for 2 years, allow the grounding line position and ice geometry to relax (Seroussi et al., 2011; Gillet-Chaulet et al., 2012). These results define the initial state of our control run and all sensitivity and sampling experiments, unless otherwise stated (See Sect. 2.3.1).

With the exception of the thermal steady state, the ISSM AIS initialization steps described above are modeled using a two-layer thin-film stress balance approximation (L1L2 Schoof and Hindmarsh, 2010; Hindmarsh, 2004). The L1L2 formulation is based on the Stokes equations, includes effects of longitudinal stresses, considers the contribution of vertical gradients to vertical shear, and assumes that bridging effects are negligible.

For the UQ experiments presented below, the continental-scale simulation must run efficiently because it is resource limited; therefore, the computationally intensive L1L2 formulation is not feasible. Alternatively, we utilize the 2D Shelfy-Stream Approximation (SSA, MacAyeal, 1989) as a surrogate stress balance model. SSA is highly efficient, computationally affordable, and proves to be a viable approximation for sampling UQ methods (Larour et al., 2012b; Schlegel et al., 2015). Additionally, it allows our simulations to run ten times faster while contributing less than 5% of uncertainty to our results (Fig. S2). More specifically, the SSA model is more efficient because it is run in 2D and therefore requires half the number of vertices. In addition, to prevent random numerical drift in UQ results our stress balance equation must converge down to a relative tolerance of at least $10^{-5}$, which in practice executes much faster with SSA. Like L1L2, SSA takes into account longitudinal stresses, making it a reasonable choice for the simulation of ice flow in areas susceptible to century-scale change, i.e. ice shelves and fast-flowing regions with low driving stresses.

## 2.2 Ocean forcing

To estimate ice shelf basal melt rates, we apply the Massachusetts Institute of Technology general circulation model (MITgcm, Marshall and Clarke (1997)) run with high horizontal ($\sim$ 9 km) and vertical (150 levels) grid spacing in a circumpolar model domain. For specific details about the method, optimization procedure, and validation strategies we refer the reader to Schodlok et al. (2016). The ocean component is coupled to a sea ice model (Losch et al., 2010) and an ice shelf module (Losch, 2008). Freezing/melting processes in the sub-ice-shelf cavity are represented by the three-equation thermodynamics of Hellmer and Olber (1989) with modifications by Jenkins et al. (2001) and implemented in the MITgcm by Losch (2008). In the ocean model, we use the steady state assumption that basal melting or freezing is balanced by glacier flow, so ice shelves are treated as rigid slabs of ice with no flexural response or change of shape. Exchanges of heat and freshwater at the base of the ice shelf are parameterized as diffusive fluxes of temperature and salinity using a constant friction velocity and the turbulent exchange coefficients of Holland and Jenkins (1999). We optimize the turbulent exchange parameters in each ice shelf cavity for the period 2004-2008 using satellite derived estimates from Rignot et al. (2013) to minimize model data differences (Schodlok et al., 2016) (See Table S1). Initial and boundary conditions are provided by the adjoint solution of the Estimating the Circulation

and Climate of the Ocean, Phase II (ECCO2) project (Menemenlis et al., 2008), and the model is run for the period 2004 to 2013 to derive a mean present-day ice shelf basal melt rate forcing.

## 2.3 ISSM forward simulations

All ISSM AIS forward simulations consist of a 100-year transient run forced by surface mass balance (SMB) from the
5 RACMO2.1 1979-2010 mean (Lenaerts et al., 2012) and by ice shelf basal melt rates from the mean 2004-2013 MITgcm simulation (See Sect. 2.2). SMB is held constant through time. Melt rates, on the other hand, are applied under the floating ice only and are updated using a nearest neighbor scheme as the modeled grounding line position evolves through time. Note that since we assume a simplified relationship for the upstream propagation melt rates, our results likely over-estimate interior melt rates, leading to overly aggressive grounding line retreat, especially in the ice sheet interior. A single 100-year SSA control
simulation launched on the NASA Advanced Supercomputing (NAS) Pleiades cluster, on 32 cpus, runs for approximately 1.5 hours. The model time step is 15.2 days, or 24 time steps per year.

### 2.3.1 Control simulation

The control simulation consists of a 100-year forward simulation as described above. During this experiment, the ice sheet interior thickens, while only minor changes occur to the grounding line position, resulting in a 0.14% increase in total ice sheet
volume above floatation over 100 years. This result is comparable to model drift reported by other continental-wide Antarctica modeling studies (i.e., Nowicki et al., 2013; Pattyn, 2017). The majority of volume change is driven by spurious transients in the model, manifesting in response to mismatches in initial boundary conditions and forcing, i.e. bedrock and ice surface elevation maps, surface ice velocities, ice shelf basal melt rates, and SMB. These errors manifest differently for any given initial condition but have limited impact on model response to anomalies in model forcing. Hence, we present each SLE result as its
simulated value minus that of its respective control simulation. This means that SLE results reflect only the model response to perturbations prescribed by the sampling experiment and neglect any present-day transient trends in AIS sea level contribution.

### 2.3.2 Complementary experiments

As the model ice front position is fixed, our results neglect changes to ice shelf back stress that would result from loss of buttressing due to ice shelf collapse. To quantify this effect, we run a collapsed ice shelf simulation. This experiment is identical
to the control simulation, except that before the start of the simulation, we remove all floating ice from the domain, such that the new ice front positions coincide with the grounding lines. Like the control run, grounding lines evolve freely, while the ice fronts are fixed.

In addition, to quantify how the bedrock affects simulation results, we repeat the initialization procedure described in Sect. 2.1, but replace the Bedmap2/MC bedrock with the Bedmap1 (Lythe and Vaughan, 2001) dataset instead.

## 2.4 Uncertainty Quantification

We use UQ analysis to investigate how uncertainties in model boundary conditions and forcing impact the Antarctic SLE contribution over 100 years, relying on the ISSM-DAKOTA framework to perform sampling experiments (Larour et al., 2012b, a; Schlegel et al., 2013, 2015; Larour and Schlegel, 2016). We focus on the forward propagation of uncertainties in ocean induced melt rates, accumulation, ice viscosity, and basal friction, and how they impact estimates of future sea level. The model domain is divided into a number of partitions, or spatial groupings of mesh vertices, and each partition is perturbed by a single constant value, randomly and independently for each 100-year simulation and for each variable being sampled. This means that each perturbation represents an instantaneous change in forcing (i.e. a step function) that persists for a century.

Values are sampled given a specified uniform distribution of uncertainty and are generated using a binned Latin Hypercube Sampling (LHS) algorithm (Swiler and Wyss, 2004), after Schlegel et al. (2015). We choose both a uniform distribution and LHS in order to emphasize sampling within the tails of the prescribed bounds, ensuring that the most extreme cases are realized by DAKOTA. In this way LHS is, in particular, an efficient tool to accomplish more statistical accuracy while running less samples (Larour et al., 2012b). Such an advantage is particularly important since, due to the computational demands of these runs, we are restricted in the number of samples we can run per experiment. The use of a uniform distribution gives no preference to any set of forcing, and assumes any value within the set bounds is equally as likely to occur. Here, our goal is to simulate plausible warming over the AIS, and due to uncertainties about the future, the forcing distribution is not straightforward to characterize; therefore, a uniform distribution is preferable to other types of sampling (i.e. a normal distribution). A normal distribution gives more weight to a forcing's mean than to its tails and may restrict our sample space. Still, it is important to note that we do not expect our choice of sampling distribution type to greatly affect our total AIS SLE contribution results, as past forward simulation sampling experiments have yielded similar diagnostic output distributions for both uniform and normal sampling (Schlegel et al., 2013).

Partitions can be determined by the user, or randomly determined using a partitioning software. For this study, we utilize both types. For the determination of random partitions, we calculate our partitions to be continuous regions of equal area, using the Chaco Software for Partitioning Graphs (Hendrickson and Leland, 1995). This allows forcing uncertainties to be weighted equally on our anisotropic mesh, as discussed by Larour et al. (2012b). We also utilize a user-defined partition configuration, that is based on geographic regions (ice sheet basins) for the purpose of constraining more plausible estimates of possible AIS SLE contribution. For instance, in a plausible realization of AIS warming, climate is not expected to change independently within small, randomly generated regions. Instead, it is likely to change on a larger more regional scale. While the use of smaller random partitions is beneficial for sampling forcing errors that are quantified spatially (i.e. with a well-defined map), such a method may underrepresent climatological-scale spatial uncertainties, or proximal correlations in forcing associated with general trends in warming. In this case, the use of many small partitions may underestimate uncertainty in model output, by artificially placing constraints on ice flow and reducing the probability of occurrence of plausible endmembers. On the other hand, during realistic climate warming, forcing is not likely to change uniformly over the entire ice sheet as atmospheric

and ocean circulation is dynamic and complex. Therefore, choosing a single partition will likely over-emphasize the output distribution's tails and overestimate its uncertainty.

Such results highlight the need for an experimental design more informed than that used in past studies (i.e. many small, random partitions) (Larour et al., 2012b; Schlegel et al., 2015). Consequentially, we are inspired to define partitions that can physically represent how climate may affect ice sheet forcing, boundary conditions, and other model input parameters. One challenge associated with this type of user-defined methodology is that in order to sample all variables simultaneously, we must choose one set of partitions that can represent spatial variability associated with climate warming for all four sampling variables. For this purpose, we define Geographic Partitions (GP), shown in white and numbered 1 through 27 in Fig. 1. These partition boundaries are informed by analysis of various high-end warming scenarios and sensitivity studies (described below in Sect. 3.2), and represent a reconciliation of inferred spatial variability for all four sampled variables. More specifically, we first separate our partitions according to ice drainage basins. This choice respects the spatially correlation of ice flow and reflects general accumulation patterns over the ice sheet (Fyke et al., 2017). Finally, in order to distinguish between areas that may be affected by the presence of liquid water, we split the basins into regions above (plateau) and below (marine) the mean ice surface elevation of 2250 m. This is the elevation line below which we assume surface melt may be present by the end of the $21^{st}$ century (i.e. Supp. Fig. in DeConto and Pollard (2016)). Such a division is also appropriate for ice shelf melt forcing, as extreme sensitivity experiments reveal that the grounding line rarely retreats beyond the present-day 2250 m surface elevation contour within the 100-year period investigated here (e.g. Fig. S5). Note that the above assumptions, which we apply to the derivation of our partitions, are informed by past studies and general expert opinion, however for subsequent studies a different partition configuration may be deemed more appropriate. As the framework allows for various strategies, the choice of partitions should be an important part of any future investigation, and the user should specifically design his or her experiment based on the scientific questions being investigated.

For this study, every UQ sampling experiment is an ensemble of 800 forward simulations in total. With each experiment, every simulation is run for 100 years with a different combination of perturbed forcing in each partition. For repeatability, we seed all experiments identically, and therefore any experiments with the same variable bounds and the same number of partitions impose identical sampling perturbations. Two-tailed statistics of output results, including distributions of total and regional SLE contribution, are computed when all samples are completed. Note that for the studies presented here, only sea level diagnostics at the end of the simulations are available. Therefore, we are restricted to use of simulation final results and do not conduct any temporally-based analyses.

We launch each ensemble on the NAS Pleiades cluster, taking advantage of the parallelized ISSM core software as well as DAKOTA's parallel management of ISSM sample simulations. Utilizing 1700 cpus, each ensemble of 800 ISSM simulations completes in less than 40 hours.

## 3 Experiments for Quantification of Uncertainty

For this study, we take two different approaches to setting range bounds for the sampled variables. In the first case, Uniform Bounds (UB), we choose bounds for each sampling variable that represent a 100-year climate warming endmember, generalized for the entire AIS. In the second approach, Informed Bounds (IB), variable bounds are specified for each GP. These regionally-based bounds are informed by past sensitivity experiments or future emission scenario model runs, such that they represent a more realistic bound for future warming for each variable. We describe the UB and IB approaches and the strategy taken for determining the bounds for each sampling variable below, in Sect. 3.1 and Sect. 3.2 respectively.

In Table 1, we list all the UB and IB UQ experiments completed for this study. This table summarizes the partitions scheme (number and type) used for each experiment as well as the range of values set for each forcing (ice shelf melt rate, accumulation, ice viscosity, and basal friction) or the geometry used (bedrock topography and ice shelves collapsed) sampled in the experiment.

The first suite of UB experiments presented in Table 1 vary only by partition configuration. We conduct these first five experiments (UB_2000, UB_500, UB_100, UB_27, and UB_1, see Fig. S3) in order to illustrate how different numbers of randomly generated equal-area partitions may affect the statistical distribution of the 100-year AIS SLE contribution. Note that we run all of these experiments using the control simulation setup described in Sect. 2.3.1. In addition, we run a UB and an IB sampling experiment using the control simulation setup and the GP configuration (UB_27GP and IB_27GP respectively) for comparison against the randomly generated partition experiments.

The remainder of the sampling experiments outlined in Table 1 are run with the GP configuration under both the UB and IB constraints and are designed to test the impact of several parameters or processes. The first two (UBCollapsed_27GP and IB-Collapsed_27GP) are run using the collapsed shelf simulation setup (see Sect. 2.3.2), in order to quantify the effect of ice shelf back stress on our results. The next two experiments (UBBedmap_27GP and IBBedmap_27GP) are run using the Bedmap1 simulation setup (see Sect. 2.3.2), in order to investigate how our current knowledge of bedrock topography impacts uncertainties in modeled sea level predictions. Finally, for the last set of experiments, we sample each of the forcing variables separately: ice shelf melt (UBMeltonly_27GP and IBMeltonly_27GP); accumulation (UBAccumonly_27GP and IBAccumonly_27GP); ice viscosity (UBViscosityonly_27GP and IBViscosityonly_27GP); and basal friction (UBFrictiononly_27GP and IBFrictiononly_27GP). This allows us to assess how each variable individually contributes to the total SLE distribution for the UB_27GP and IB_27GP sampling experiments.

### 3.1 Uniform Bounds

For the UB experiment, we follow a typical sensitivity approach to ice sheet model projections. Variable bounds do not vary regionally, and are chosen largely to reflect changes in forcing that could result from large-scale warming of the climate over the next century. Here, we focus on exposing the system to a large range of plausible climate realizations to evaluate the model stability and feedbacks.

**Table 1.** List of sampling experiments and associated ranges of sampled variables for the UB and IB experiments. Here, we report all UB. IB are marked as 'regional' and are displayed in Table 2. Partitions are randomly generated, unless noted by GP, in which case they are geographically-based. For ice shelf basal melt, the maximum bounds are given as melt multipliers, while all other variable bounds are given in percentage change. Minimum melt bounds within a partition are not represented here because they not represented by a uniform value (See Table 2 and Sect. 3.1).

| Name | Number of partitions | Accumulation Min/Max (%) | Melting rate Max Multiplier | Basal friction Min/Max (%) | Ice viscosity Min/Max (%) | Shelf collapse | Bedrock topography |
|---|---|---|---|---|---|---|---|
| UB_2000 | 2000 | $-50/+100$ | $\times 10$ | $-60/+0.01$ | $-40/+0.01$ | No | Bedmap2/MC |
| UB_500 | 500 | $-50/+100$ | $\times 10$ | $-60/+0.01$ | $-40/+0.01$ | No | Bedmap2/MC |
| UB_100 | 100 | $-50/+100$ | $\times 10$ | $-60/+0.01$ | $-40/+0.01$ | No | Bedmap2/MC |
| UB_27 | 27 | $-50/+100$ | $\times 10$ | $-60/+0.01$ | $-40/+0.01$ | No | Bedmap2/MC |
| UB_1 | 1 | $-50/+100$ | $\times 10$ | $-60/+0.01$ | $-40/+0.01$ | No | Bedmap2/MC |
| UB_27GP | 27 GP | $-50/+100$ | $\times 10$ | $-60/+0.01$ | $-40/+0.01$ | No | Bedmap2/MC |
| IB_27GP | 27 GP | regional | regional | regional | regional | No | Bedmap2/MC |
| UBCollapsed_27GP | 27 GP | $-50/+100$ | $\times 10$ | $-60/+0.01$ | $-40/+0.01$ | Yes | Bedmap2/MC |
| IBCollapsed_27GP | 27 GP | regional | regional | regional | regional | Yes | Bedmap2/MC |
| UBBedmap_27GP | 27 GP | $-50/+100$ | $\times 10$ | $-60/+0.01$ | $-40/+0.01$ | No | Bedmap1 |
| IBBedmap_27GP | 27 GP | regional | regional | regional | regional | No | Bedmap1 |
| UBMeltonly_27GP | 27 GP | fixed | $\times 10$ | fixed | fixed | No | Bedmap2/MC |
| IBMeltonly_27GP | 27 GP | fixed | regional | fixed | fixed | No | Bedmap2/MC |
| UBAccumonly_27GP | 27 GP | $-50/+100$ | fixed | fixed | fixed | No | Bedmap2/MC |
| IBAccumonly_27GP | 27 GP | regional | fixed | fixed | fixed | No | Bedmap2/MC |
| UBViscosityonly_27GP | 27 GP | fixed | fixed | $-60/+0.01$ | fixed | No | Bedmap2/MC |
| IBViscosityonly_27GP | 27 GP | fixed | fixed | regional | fixed | No | Bedmap2/MC |
| UBFrictiononly_27GP | 27 GP | fixed | fixed | fixed | $-40/+0.01$ | No | Bedmap2/MC |
| IBFrictiononly_27GP | 27 GP | fixed | fixed | fixed | regional | No | Bedmap2/MC |

For accumulation, we set the minimum bound to a $-50\%$ change and the upper bound to a $+100\%$ change in the control run accumulation forcing. The accumulation minimum bound represents the historical maximum annual variability in SMB over the last 200 years, as estimated from ice cores (Frezzotti et al., 2013). The upper bound is equivalent to the precipitation change expected due to an increase in surface temperature of about $10°C$ , in accordance with the Clausius-Clapeyron relation (for current mean temperatures over the AIS). This value is representative of estimated accumulation changes in high elevation areas of East Antarctica, but is likely an overestimate in the West AIS (Frieler et al., 2015).

For ice viscosity, we set the minimum bound to a $-40\%$ change in ice viscosity used in the control run, to account for an extreme decrease in ice viscosity due to cryo-hydrologic warming from increases in surface temperature and surface liquid water (rain and/or surface melt) (Schlegel et al., 2015). In the most extreme case, this assumes that surface ice temperatures reach the melting point of ice during the summer (from $5°C$ warming along the margins to a $30°C$ warming in the deep interior of the continent). In this case, surface water could percolate through the ice column and refreezes at depth, rapidly altering the ice thermal regime (Phillips et al., 2010, 2013). Where required, minimum bounds are restricted in order to ensure that all values remain above the viscosity of ice at melting point (Cuffey and Paterson, 2010). On floating ice, this bound also represents a change to the ice shelf rheology (i.e., damage) due to the percolation of surface melt into the floating ice and structural weakening (Borstad et al., 2012). As we do not consider increases to ice viscosity to be realistic, we set the maximum bound to the control run value plus 0.01%.

For basal friction, we set the lower bound to a $-60\%$ change in the values used in the control run, which is equivalent to the SeaRISE experiment S2 (Nowicki et al., 2013), i.e., a 2.5 times increase in sliding. We set the basal friction upper bound to the control run value plus 0.01%.

For ice shelf basal melt rates, we do not set the lower bounds as a uniform change, but instead consider the interannual variability within each partition by setting it to the regional area-averaged percent difference between the mean annual melt rate and the minimum annual melt rate. This change is generally small, i.e. for the GP configuration, the largest melt rate reduction is $0.1$ m $y^{-1}$. For the upper bounds, we choose a multiplication factor of 10 times the control run melt rate, which for Pine Island Glacier represents a grounding line potential temperature increase larger than $10°C$ (considering the temperature dependency of Rignot and Jacobs (2002)). While this factor represents a most extreme ocean forcing for warm ice shelves (e.g. Amundsen Sea), it is important to note that it also underestimates potential warming in others (e.g., cold shelves like Filchner-Ronne Ice Shelf, Hellmer et al. (2012)). This dichotomy illustrates the difficulty in designing an appropriate ice shelf melt rate perturbation experiment for Antarctica, as typically, a single offset or multiplier value is chosen to represent a generalized warming experiment.

## 3.2 Informed Bounds

In the IB experiment, we refine GMSL projections over the coming century, by defining sampling bounds that are regionally informed by model projections or targeted model sensitivity studies. All IB experiments are conducted on 27 GP, and sampling variable bound ranges are determined separately for each partition.

IB lower bounds for ice shelf melt rates are the same as UB. Upper bounds for ice shelf melt rates are determined from MITgcm idealized sensitivity experiments (see Appendix Sect. A). All bounds for IB ice shelf melt rates are expressed as multiplication factors of the control run melt rate values (Table 2).

We base accumulation bounds on results from the Coupled Model Intercomparison Project Phase 5 (CMIP5) (Pachauri et al., 2014; Church et al., 2013), using results from the 80 publicly available RCP8.5 emissions scenario (Taylor et al., 2012) atmospheric simulations to characterize possible future spread in AIS precipitation change over the next century due to high greenhouse gas emission warming. Upper and lower IB are set respectively as the maximum and minimum percent simulated accumulation change, with consideration to output from the 80 CMIP5 models, between the mean of years 2006-2010 and 2096-2100.

Informing bounds for ice viscosity for the next century is more challenging, as it is difficult to assess how changes in climate will impact the thermal properties of the ice. We set a minimal base estimate of ice viscosity bounds equal to $\pm 5\%$ of the control run value since changes in surface temperatures propagate slowly through the ice column, and would not be expected to significantly alter ice temperatures on the timescales considered here (Seroussi et al., 2013). To account for cryo-hydrologic warming and any ice dynamic changes due to increases in surface melt, we decrease the lower bounds in low elevation regions where liquid water is expected to be present by the end of the $21^{st}$ century (i.e. regions below 2250 m). The new lower bounds are informed by CMIP5 ice sheet model simulations under the RCP8.5 scenario (See e.g. Supp. Fig. in DeConto and Pollard (2016)) such that, depending on the amount of melt water simulated within a specific partition by year 2100, ice viscosity lower bounds range from $-10\%$ to $-6\%$ change.

For basal friction, uncertainties are not well defined, and associated processes are part of an active field of research. For the Greenland Ice Sheet, surface runoff reaches the base of the ice sheet, and basal sliding increases or decreases, depending on the type of subglacial drainage system developed (Bell, 2008; Sundal et al., 2011). In Antarctica, limited amount of supraglacial water is available to impact basal conditions, as most subglacial water is linked to basal heating (due to geothermal heat flux and frictional heat, i.e. Pattyn (2010); Seroussi et al. (2017a)) and is not directly affected by climate change. Consequentially, model errors in basal sliding can be mainly attributed to uncertainties in the inferred basal friction. We characterize this uncertainty by comparing the inferred basal friction from ISSM AIS models initialized with various spatial resolutions and with different bedrock topography maps. Results indicate that basal drag coefficient uncertainties (Larour et al. (2012c), eq. 19) are about $\pm 15\%$ in East Antarctica, and reach a maximum of $\pm 25\%$ in West Antarctica. Translated into IB for basal friction in accordance with ISSM's friction law (Larour et al., 2012c), these values are equivalent to a $-28\%$ and $-44\%$ change for the lower bounds and a $+32\%$ and $+56\%$ change for the upper bounds in East and West AIS respectively. Note that these bounds may be an underestimate of the actual uncertainty in basal friction, as they capture only variability within the Budd-type friction law used here. Consequentially, our results do not consider any uncertainty in basal friction that may be sourced in the choice of the basal friction law itself (Brondex et al., 2017; Pattyn, 2017) .

**Table 2.** For the IB experiments, associated range of sampled variables for each GP (Fig. 1) . The last two columns list the mean upper bound ice shelf melt rates forced for each marginal partition during the sampling experiments, for IB and UB respectively. The mean upper bound melt rate is equal to the maximum melt rate multiplier times the control ice shelf melt rate values, area averaged over the initialization ice shelf area within each margin partition.

| Geographic partition number | Accumulation Min/Max (%) | Melting rate Min/Max Multiplier | Basal friction Min/Max (%) | Ice viscosity Min/Max (%) | Mean Upper Melt rate IB (m y$^{-1}$) | Mean Upper Melt rate UB (m y$^{-1}$) |
|---|---|---|---|---|---|---|
| 1 | $-5.52/+49.96$ | $\times 0.65/\times 18$ | $-43.75/+56.25$ | $-10/+5$ | 16.94 | 9.4 |
| 2 | $-49.72/+66.67$ | $\times 0.99/\times 3.5$ | $-43.75/+56.25$ | $-8/+5$ | 9.77 | 27.7 |
| 3 | $-30.70/+57.19$ | $\times 0.99/\times 6.2$ | $-43.75/+56.25$ | $-7/+5$ | 20.28 | 32.8 |
| 4 | $-29.50/+66.76$ | $\times 0.99/\times 1.8$ | $-43.75/+56.25$ | $-8/+5$ | 20.83 | 118.3 |
| 5 | $-58.34/+71.60$ | $\times 0.99/\times 2.7$ | $-43.75/+56.25$ | $-6/+5$ | 9.48 | 34.8 |
| 6 | $-5.00/+54.77$ | $\times 0.75/\times 40$ | $-43.75/+56.25$ | $-8/+5$ | 34.4 | 8.6 |
| 7 | $-8.31/+87.60$ | $\times 0.35/\times 63.2$ | $-43.75/+56.25$ | $-6.5/+5$ | 23.37 | 3.7 |
| 8 | $-10.82/+101.70$ | $\times 0.98/\times 31$ | $-43.75/+56.25$ | $-6.5/+5$ | 12.4 | 4 |
| 9 | $-10.23/+124.25$ | $\times 0.83/\times 45.7$ | $-43.75/+56.25$ | $-6/+5$ | 17.38 | 3.8 |
| 10 | $-86.58/+78.68$ | $\times 0.41/\times 15.1$ | $-43.75/+56.25$ | $-8.5/+5$ | 23.56 | 15.6 |
| 11 | $-90.72/+89.08$ | $\times 0.22/\times 15.1$ | $-43.75/+56.25$ | $-6/+5$ | 23.56 | 15.6 |
| 12 | $-47.63/+111.00$ | $\times 0.93/\times 5$ | $-27.75/+32.25$ | $-5/+5$ | 9.75 | 19.5 |
| 13 | $-23.86/+92.58$ | $\times 0.94/\times 62.9$ | $-27.75/+32.25$ | $-7.5/+5$ | 39.63 | 6.3 |
| 14 | $-5.00/+104.42$ | $\times 0.99/\times 62.9$ | $-27.75/+32.25$ | $-5/+5$ | – | – |
| 15 | $-5.00/+91.95$ | $\times 0.99/\times 31$ | $-27.75/+32.25$ | $-5/+5$ | – | – |
| 16 | $-46.15/+107.71$ | $\times 0.99/\times 15.1$ | $-27.75/+32.25$ | $-5/+5$ | – | – |
| 17 | $-5.00/+105.93$ | $\times 0.99/\times 20$ | $-27.75/+32.25$ | $-5/+5$ | – | – |
| 18 | $-25.57/+86.78$ | $\times 0.99/\times 11.8$ | $-27.75/+32.25$ | $-5/+5$ | – | – |
| 19 | $-73.31/+97.52$ | $\times 0.96/\times 11.8$ | $-27.75/+32.25$ | $-6/+5$ | 29.51 | 25.1 |
| 20 | $-59.29/+154.64$ | $\times 0.99/\times 100$ | $-27.75/+32.25$ | $-5/+5$ | – | – |
| 21 | $-5.00/+111.61$ | $\times 0.99/\times 51$ | $-27.75/+32.25$ | $-5/+5$ | – | – |
| 22 | $-8.12/+99.33$ | $\times 0.99/\times 172.2$ | $-27.75/+32.25$ | $-5/+5$ | – | – |
| 23 | $-5.00/+120.22$ | $\times 0.99/\times 7$ | $-27.75/+32.25$ | $-5/+5$ | – | – |
| 24 | $-53.61/+122.85$ | $\times 0.99/\times 100$ | $-27.75/+32.25$ | $-6.5/+5$ | 89 | 8.9 |
| 25 | $-32.24/+146.85$ | $\times 0.87/\times 51$ | $-27.75/+32.25$ | $-6.5/+5$ | 32.64 | 6.4 |
| 26 | $-12.45/+141.01$ | $\times 0.88/\times 100$ | $-27.75/+32.25$ | $-6.5/+5$ | 156 | 15.6 |
| 27 | $-19.23/+139.85$ | $\times 0.99/\times 7$ | $-27.75/+32.25$ | $-6.5/+5$ | 26.84 | 38.4 |

## 4    Results

The goal of this study is to use sampling analysis to quantify the uncertainty in simulated SLE contribution from the Antarctic Ice Sheet over a 100-year period. We also investigate how this uncertainty varies regionally. SLE contribution is determined by the change in ice volume above floatation during the 100-year simulation period, compared to the change in ice volume above floatation during the 100-year control run.

Our UQ results are presented in a number of formats. Distribution plots represent the probability density function (PDF) for the frequency of SLE contribution from Antarctica for the 800 independent random samples at the end of a 100-year long forward simulation. In addition, all AIS continental SLE contribution results are summarized in Table 3. To aid in regional analysis, bar plots represent the mean and standard deviation of SLE contribution from the sample simulations for every GP.

### 4.1    Impact of partition choice

For this type of resource-limited UQ study, experimental design is likely to impact the distribution and magnitude of the AIS SLE contribution, particularly the choice of partition configuration. To investigate this effect, we run a suite of UB experiments (such that for each of the four variables synchronously sampled, the sampling bounds are the same for every partition), altering only the partition configurations (See Fig. S3). The experiments consist of a single partition sampling, plus four experiments that are run with various numbers of randomly generated partitions (Table 1). Additionally, we include an experiment with GP partitioning (See Sect. 2.4).

In Fig. 2, we compare the results of these first six sampling experiments. Overall, we find that the experiments vary in uncertainty and, as Fig. 2 illustrates, a larger number of partitions results in a smaller overall uncertainty. The single partition experiment (UB_1) has a distinctly larger uncertainty and reveals a SLE potential almost double that of the randomly generated partition experiments, reaching a maximum 95% confidence interval of about 1.4 m SLE (Table 3, UB_27, UB_100, UB_500, UB_2000). Technically, if enough samples are run for each experiment, they should all have equivalent endmembers because they are all restricted by the same continental-scale bounds. Yet, results suggest that increasing the number of partitions decreases the probability of endmember occurrence, and we find that the standard deviation degrades when the number of partitions is increased (Fig. 2 and Table 3). Statistically, this result could be biased since we are restricted to 800 samples for each experiment and we may not realize the entire random sample space for those experiments with a larger number of partitions. However, analysis of additional experiments with increased numbers of samples (not shown) suggests that with LHS uniform sampling, 800 samples are adequate and that the resulting PDF curves shown here are robust. That is, our UB and IB PDF curves converge within 800 samples, with PDF means and spreads varying by less than 0.6% beyond 800 samples. Indeed, the resulting spread is partly dictated by the fact that, while the number of partitions increases, the PDF curves for the total mass forcing (specifically the total continental SMB and total basal melt) tend to become more normally distributed and begin to narrow (a phenomenon dictated by the Law of Large Numbers and the Central Limit Theorem). After further analysis, we also find that such variations in spread can be contributed to dynamic feedbacks between the partitions. For instance, while UB_1 assumes that forcing of climatically driven parameters would change uniformly over the entire ice sheet, UB_2000 assumes

**Table 3.** SLE (m) and associated statistics resulting from 800 simulations for each of the experiments described in Table 1.

| Name | Mean SLE (m) | Standard deviation (m) | Min 95% confidence interval (m) | Max 95% confidence interval (m) |
|---|---|---|---|---|
| UB_2000 | 0.6151 | 0.1341 | 0.358 | 0.779 |
| UB_500 | 0.5963 | 0.1489 | 0.317 | 0.776 |
| UB_100 | 0.5285 | 0.1584 | 0.252 | 0.758 |
| UB_27 | 0.4843 | 0.1938 | 0.162 | 0.766 |
| UB_1 | 0.6566 | 0.4692 | $-0.124$ | 1.386 |
| UB_27GP | 0.6282 | 0.1940 | 0.295 | 0.904 |
| IB_27GP | 0.3292 | 0.1019 | 0.152 | 0.495 |
| UBCollapsed_27GP | 0.9932 | 0.1744 | 0.697 | 1.235 |
| IBCollapsed_27GP | 0.5363 | 0.0801 | 0.407 | 0.676 |
| UBBedmap_27GP | 0.5136 | 0.1877 | 0.163 | 0.767 |
| IBBedmap_27GP | 0.1845 | 0.08834 | 0.034 | 0.331 |
| UBMeltonly_27GP | 0.5141 | 0.1396 | 0.261 | 0.700 |
| IBMeltonly_27GP | 0.4847 | 0.07668 | 0.340 | 0.590 |
| UBAccumonly_27GP | $-0.1561$ | 0.06220 | $-0.265$ | $-0.056$ |
| IBAccumonly_27GP | $-0.1994$ | 0.05704 | $-0.299$ | $-0.109$ |
| UBViscosityonly_27GP | 0.03597 | 0.009540 | 0.021 | 0.052 |
| IBViscosityonly_27GP | $-0.0005772$ | 0.004608 | $-0.009$ | 0.007 |
| UBFrictiononly_27GP | 0.1120 | 0.02250 | 0.074 | 0.150 |
| IBFrictiononly_27GP | 0.003129 | 0.01680 | $-0.024$ | 0.032 |

minimal spatially proximate correlation. That means that with many partitions it is less likely that all partitions of the ice sheet will change together to promote mass gain or mass loss. Specifically, for a particular sample simulation, it is more likely that at least one partition within a drainage basin is promoting ice gain while the rest of the basin partitions are promoting ice loss. In this case, the one partition promoting ice gain may stall or counteract mass loss within the entire basin, making the occurrence of an endmember sample result even more rare as the number of sampling partitions increases.

Table 3 and Fig. 2 illustrate the effect of choosing partitions that promote spatial correlation (i.e. correlation within ice flow basin and by surface elevation), in contrast to the randomly generated partition distributions. The distributions of UB_27

with UB_27GP are almost equivalent in standard deviation and frequency, but the UB_27GP is shifted approximately 0.14 m SLE to the right, indicating that the GP run is more likely to lose more mass than the run with randomly generated partitions. The GP configuration respects ice flow boundaries on the 100-year timescale considered here and therefore, does not bisect large glaciers or ice streams, as the randomly generated partitions are likely to do (e.g., Pine Island Glacier, Fig. S3). Within the GP configuration, large glaciers and ice streams in the same basin change synchronously (uniformly with respect to each independently sampled variable), which is likely to promote glacier instability and ice mass loss. Consequently, we choose to utilize the GP configuration for all other UQ experiments presented in this manuscript, as sampling with this configuration is representative of physically based correlations within the ice sheet system not captured by randomly generated partitions.

One important characteristic of the partition experiment distributions is that, for the random (equal area) experiments, the distribution mean shifts to the left with the use of less partitions (Fig. 2). This phenomenon occurs in response to the sampling of accumulation, which is directly responsible for dictating the amount of mass that is added to the ice sheet (Fig. 3). The less the number of partitions, the larger the partition regions, and the more likely increases in accumulation will affect a significant portion of the AIS. Forcing a larger ice sheet area with significant increases in accumulation (up to 100%) results in mass storage, encouraging the ice sheet to grow and negatively affecting SLE contribution. Therefore, when running with a smaller number of larger sized partitions, the simulations are more likely to gain mass, resulting in a SLE PDF shift to the left.

## 4.2 Uniform bounds experiments

For the UB forcing experiments run with greater than one partition, we find that the resulting distributions are distinctively bi-modal (Fig. 2). We conduct sampling experiments independently on each variable (UBMeltonly_27GP, UBAccumonly_27GP, UBViscosityonly_27GP, UBFrictiononly_27GP) in order to gain insight into the effects of each forcing on our resulting distributions. Analysis of an experiment forced only with ice shelf melt (UBMeltonly_27GP), isolates this variable as the only source with a bimodal distribution (Fig. 3a and Fig. S4). By sampling ice shelf melt rates alone, we remove the feedbacks from the other sampled variables. This feedback is non-trivial, as evidenced by the difference between the variable summed and the combined sampling curve in Fig. 3a. It is clear that for our UB runs, ice shelf melt is responsible for the majority of the spread (Table 3 and Fig. 3a) as well as for the bimodal behavior of the UB distributions (Fig. S4).

Additional regional analysis pinpoints the Amundsen Sea (GP 4) as the dominant source of the AIS uncertainty (Fig. 4), suggesting that the large ice shelf melt rates forced in this area are almost entirely responsible for the spread of the total AIS PDF (Fig. 4). These results agree with those from past model sensitivity experiments and projection-based AIS warming scenario simulations that have reported complex grounding line and ice dynamic responses within the Amundsen Sea (e.g., Cornford et al., 2015; Ritz et al., 2015). Under the UB experiment, we find that the mean SLE contribution from Amundsen Sea is 297 mm, accounting for about 47% of the continental AIS mean SLE contribution and about 89% of its uncertainty (difference between the maximum and minimum 95% confidence intervals). The forcing of ice shelf melt rates alone accounts for 82% of the Amundsen Sea's mean SLE response and about 72% of its uncertainty (Fig. 4).

Other regions responsible for SLE contribution are GP 27 (Wilkes Land coast, including Totten Glacier and Moscow University Ice Shelf) and GP 7 (the Ronne Ice Shelf basin, particularly Institute and Möller Ice Streams) (Fig. 4). Single sensitivity

runs forced with endmember warming bounds indicate that western Ross Ice Shelf / Siple Coast (GP 10) is also expected to be a larger contributor to SLE on timescales longer than considered here (∼200 years, Fig. S6).

## 4.3 Informed Bounds experiments

Unlike the UB experiments, the IB experiments exemplify a less complex unimodal distribution. Indeed, the changes in sampling bounds compared to the control run vary regionally for each parameter in the IB experiments. In contrast to the UB experiments, the IB encompass realizations that are foreseeable according to the literature, sensitivity experiments, and CMIP5 ensemble mean projections for the end of the century (See Sect. 3.2). Under these restrictions, the IB experiments results are still dominated by the ice shelf melt rate forcing (Fig. 3). Also similar to the UB experiments, the effects of increased ice shelf melt rates are mitigated by increases in accumulation (the most important variable after melt rates). Basal friction and ice viscosity play minor roles in the resulting SLE distributions, as over the 100-year period we do not expect surface runoff to increase substantially enough to affect these parameters on a basin scale. In the UB experiments, where Amundsen Sea ice shelf melt is the most significant contributor, the maximum bounding melt rate is equivalent to a 10 times multiplier of the control melt rates. Overall, we do not consider this value realistic for the Amundsen Sea sector (GP 4, Fig. 1) over the next century, especially since melt rates in this area are already substantially greater than melt rates for the rest of the AIS (i.e. warm ocean waters currently reach the largest glaciers in the region).

In contrast, for the IB experiments, the estimated maximum value is substantially less: a multiplier of 1.8 times the current melt rates (Table 2). The mean SLE contribution from the Amundsen Sea due to our sampled forcing is 6 mm in 100 years (Fig. 4), which is equivalent to the mass loss acceleration already observed in the West AIS during last decade (Shepherd et al., 2018). However, the most extreme IB simulations reveal that an addition 66 mm of SLE contribution from the Amundsen Sea is plausible in response to 100 years of perturbed forcing (Fig. 4). This response represents an acceleration of about three times the present day rate of mass loss in this region (Medley et al., 2014). We find that with respect to the total AIS SLE potential contribution under the IB experiment, the Amundsen Sea is a relatively minor contributor to future sea level rise (Fig. 4). It is conceivable that warm waters reach the continental shelf, heating the ocean waters beneath traditionally cold ice shelves, and significantly increasing the ice shelf melt potential within these current regions of low melt rates (Hellmer et al., 2012). Our ocean model sensitivities studies (Sect. 3.2 and Appendix Sect. A) reveal a similar warming potential beneath the Ronne Ice Shelf, resulting in relatively large endmember melt rates for our IB experiments (Table 2). As a result, we find that the most significant potential contributors are ice streams feeding Ronne Ice Shelf (GP 7), particularly the Evans and Rutford Ice Streams (Fig. S5c) as well as the Institute and Möller Ice Streams, together resulting in a 161 mm mean SLE contribution (∼50% of the total AIS contribution). The SLE contributions from these ice streams are dominated by grounding line retreat forced by prescribed ice shelf melt rates, and therefore these results are direct consequences of the large upper bound multiplier chosen for GP 7 (Table 2). In this case, ice shelf melt alone is responsible for approximately all of the mean response for this region and 82% of its uncertainty (Fig. 4).

Other contributing regions include GP 8 (Filchner Ice Shelf basin: Slessor and Recovery Ice Streams), GP 10 (western Ross Ice Shelf: Siple Coast), GP 13 (including Jutulstraumen Glacier, Fimbul and Lazarev Ice Shelves), and GP 25 (Amery Ice

Shelf basin) (Fig. 4). Note that results from single simulation sensitivity runs forced with IB endmember warming bounds indicate that under the IB experiment the Amundsen Sea may be a more significant contributor to SLE on a timescale beyond the 100 years considered here (Fig. S6), as suggested by other regionally-based AIS modeling experiments (Cornford et al., 2015). Of additional interest are GP 26 and GP 27. Within these regions, ice shelf melt IB allow for potential SLE contributions
similar to those listed above, however, accumulation bounds in these regions are substantial enough to counterbalance ice that is potentially lost to elevated ice shelf basal melt rates (Fig. 4) (up to an upper bound of 140% increase in accumulation, Table 2). This leads to large uncertainties within these regions around a lower mean SLE contribution that would be much higher if melt were sampled alone.

## 4.4 Impact of ice shelf collapse

As our simulations do not include migrating ice fronts, and therefore subsequent reduction in ice shelf buttressing, they may underestimate future sea level rise. To quantify the effect of back stress provided by the current ice shelves, we conduct sampling experiments on a control run that instantaneously loses all of its current ice shelves at the beginning of the run (See Sect. 2.3.2). While the removal of ice shelves does not affect SLE contribution directly (ice shelves are floating), it eliminates any buttressing of the interior ice that is provided by the current ice shelves.

Fig. 5a shows the UBCollapsed_27GP and IBCollapsed_27GP experiments' SLE PDFs as well as the UB_27GP and IB_27GP experiments' SLE PDFs. Generally, we find that the effect of collapsing AIS ice shelves creates an additional SLE contribution of 0.37 and 0.32 m for UB and IB experiments respectively. In addition, experiments including initial ice shelf collapse show a small narrowing in their uncertainty ranges (Fig. 5a and Table 3). It is clear that the regions most affected are those with the largest ice shelves, and therefore the most buttressing. The areas significantly affected include GP 7 and 8 (the
Filchner-Ronne basin), GP 10 (western Ross Ice Shelf), and GP 13 (Dronning Maud Land coast, including Fimbul and Lazarev Ice Shelves). Other sensitive regions of minor note include GP 5, GP 25, and GP 26 (Fig. 6).

For the regions noted above, we find that the standard deviation is reduced for the collapsed ice shelf experiment (Fig. 6). This is indicative of the loss of a number of lower bound sample runs in which grounding line retreat is mitigated. Indeed, once the ice shelves are instantaneously removed, these areas are already committed to SLE contribution in response to the removal
of present day ice shelf buttressing. This is particularly pertinent in the case of the large cold Filchner-Ronne and Ross Ice Shelves, where the collapsed ice shelf SLE contribution is comparable between the UB and IB experiments (means within $1\sigma$), even though their melt rate upper bounds have drastically different values (Table 2). Such results highlight the importance of the role of ice shelf buttressing in the stability of the glaciers that feed into cold ice shelf regions. Indeed, our results represent a lower bound on the effects of future ice shelf collapse, since ice margins do not migrate through time.

## 4.5 Impact of bedrock topography

Another significant boundary condition that dictates stability within a glacier system is its bedrock topography. To illustrate the importance of improved resolution of bedrock on century-scale projections, we repeat our spin up and GP sampling experiments using Bedmap1, instead of the current state-of-the-art bedrock topography (Bedmap2 refined with mass conservation

techniques), to define the model bedrock (See Sect. 2.1 and Sect. 2.3.2). In Fig. 5b, we compare the results of UBBedmap_27GP and IBBedmap_27GP with UB_27GP and IB_27GP. We find that for the continental AIS SLE contribution, the mean values for the UB UBBedmap_27GP and UB_27GP experiments do not differ significantly, while for the IB experiment, the IBBedmap_27GP and IB_27GP means differ beyond a standard deviation (Fig. 5 and Table 3). Overall, the improvement from Bedmap1 to Bedmap2/MC results in a right shift of 0.11 m for the UB experiments and 0.14 m for the IB experiments. Both experiments reveal a slight widening of uncertainty bounds with the change from Bedmap1 to Bedmap2/MC (with an increase in uncertainty of 18% and 1% for IB and UB respectively), suggesting that the improved bedrock topography results in an increase in the sample space of possible grounding line retreat realizations as well as a widening of the tails (Fig. 5b).

Regional analysis reveals that the quality of the bedrock topography affects the rate of retreat of specific glaciers, depending on the forcing. For instance, in the UBBedmap_27GP experiment, the Amundsen Sea contributes 15% (or 44 mm) more SLE than does the UB_27GP experiment (Fig. 6). Yet, for the UB Bedmap1 experiments overall, we find that regional decreases in mass loss (particularly in the East AIS) outweigh the Amundsen Sea mass loss increases. The area with the largest mass loss decrease due to the Bedmap1 topography is GP 27 (Wilkes Land coast, including Moscow University Ice Shelf and Totten Glacier, Fig. 6), contributing a mean of 26 mm and 46 mm less to GMSL over the 100-year simulation for IB and UB respectively. For the IB experiment, in all regions IBBedmap_27GP contributes less SLE than does IB_27GP (Fig. 6). In particular, GP 26 and GP 27 are affected by the different bedrock topography maps, as is the largest SLE contributor, GP 7 (Ronne Ice Shelf) (Fig. 6).

For key areas noted above, we illustrate the differences between Bedmap1 and Bedmap2/MC and the final grounding line positions for endmember single forward simulations in Fig. S5. The differences in final grounding line positions demonstrate the effect of improved bedrock detail within major outlet glaciers and emphasize how different bedrock topographies alter ice sheet model projections. Indeed, with improved detail in the bedrock topography, the simulation of grounding line migration and glacier retreat in turn acquire enhanced detail and complexity.

## 5    Discussion

Model projections of future sea level rise are inherently uncertain and depend largely on the model boundary conditions and climate forcing. Ice sheet models are specifically designed to capture the complex dynamic responses to ocean and atmospheric change, compounding the difficulty of pinpointing the potential sources of model uncertainty. Here, we uniquely take advantage of the ISSM-DAKOTA UQ framework, which allows for a relatively high spatial resolution near shear margins and the grounding line, a feature advantageous for the physical modeling of ice flow and grounding line retreat and an improved representation of bedrock topography. Because ISSM also has the option of running with various stress balance approximations, the experimental design described here takes advantage of a two dimensional representation of ice flow that reasonably replicates results from a more complex stress balance model (Fig. S2 and Sect. 2.3.1). Thus, the ISSM-DAKOTA highly parallelized, high-resolution UQ framework allows for continental and regional statistical assessment of AIS dynamic mass change.

Using this unique framework, we characterize model uncertainty by capturing a large sample space of potential century-scale AIS SLE contributions. Despite the wide range of forcing, we find that the ice sheet's mass balance response is normally distributed, except when locally extreme ocean forcing is applied. Specifically, forcing with extreme ice shelf melt rate bounds results in a total AIS SLE contribution distribution that is bimodal, which illustrates the dynamic complexity of grounding line response. Furthermore, we find grounding line retreat is locally enhanced when the simulation is forced by a simultaneous reduction in basal friction, ice viscosity, or accumulation (i.e. a mean enhancement of 12% for IB and 19% for UB experiments when all variables are combined, Fig. 3), which is a source of added complexity in our results. Indeed, the response of the grounding line position to increased ice shelf basal melt rate forcing is strongly dependent on the applied melt rate as well as the local conditions, particularly bedrock geometry and ice-shelf buttressing, which determine grounding line stability. This means that the resulting SLE PDFs are not only a ramification of forcing, boundary conditions, and input parameters, but also a combined consequence of various regional responses of many individual glaciers.

These results suggest that, in particular, it is most important to reduce uncertainty in the representation of grounding line migration and in ice shelf basal melt rate forcing. In the experiments presented here, we utilize an optimized reconstruction of historical melt rates from an ocean model run at a horizontal grid spacing of 9 km, and for the ice sheet model areas of floating ice are restricted to a horizontal resolution of 8km or finer with coarser grid spacing upstream of the grounding line. For the ocean model, the resulting grid is an improvement over past studies in terms of resolving the circulation beneath key ice shelves (Schodlok et al., 2016), but those areas with smaller outlet glaciers and ice shelves still require finer grid spacing. These areas include Enderby Land (GP 24) and George V Land / Davis Sea sector (GP 26), whose melt rates remain highly uncertain (Table 2, Table S1, and Appendix Sect. A). For the ice sheet model, the resolution used here is also improved over previous continental scale studies (Ritz et al., 2015; DeConto and Pollard, 2016; Pattyn, 2017), yet in order to properly model grounding line migration, the ice sheet model should still be run with a horizontal resolution finer than 8 km near the present day grounding line (Durand et al., 2009; Gladstone et al., 2010; Seroussi et al., 2014a). Due to computational restrictions, the horizontal mesh resolution of both the ocean and ice sheet model used here are restricted to values coarser than desired, especially in the interior, where the grounding line migrates into regions with resolutions even lower than 8 km (Fig. S1 and Fig. S5). Currently, finer resolutions can be accomplished with the use of regional scale models; but ideally, future improvements would include coupling of highly resolved continental scale ocean and ice sheet models, in lieu of parameterizations, in order to best represent the evolution of melt rates as grounding lines migrate in the future. A coupled ocean-ice sheet model capable of resolving important topographical features may improve the estimates of melt rates beneath ice shelves as grounding lines retreat by simulating the evolution of the shape of the cavity between the ice shelves and the bedrock topography. Indeed, by representing the physical changes in ocean circulation patterns beneath new ice shelves, the models are expected to produce more credible estimates of ice shelf melt rates (De Rydt and Gudmundsson, 2016; Seroussi et al., 2017b).

In order to better assess the impact of mesh resolution on the uncoupled ice sheet model simulations used here, we conducted sensitivity experiments on a regional model of Thwaites Glacier (Seroussi et al., 2017b). Results suggest that the ISSM mesh resolutions are sufficient to capture the general behavior that the same model would produce with a four times finer spatial resolution. Additional experiments conducted on a lower resolution ISSM AIS (where the finest model spatial resolution is 3

km instead of the 1 km used in this study) illustrate how model resolution can affect the continental AIS model response in a complex way (Fig. S4). More specifically, we find that the lower resolution model results reveal a slight decrease in SLE AIS contribution for the IB experiments and an increase for the UB experiments. For the UB experiment, this increase is a direct consequence of how mesh resolution affects the model response to ice shelf melt rates (Fig. S4), in agreement with results from Pattyn (2017) on a 100-year timescale using coarser simulation meshes. Results on the basin scale (not shown) reveal that the difference between the higher and lower mesh resolution models are distinctively attributed to the dynamic behavior of the ice streams feeding the Filchner-Ronne Ice Shelves, the Amundsen Sea sector, and Wilkes Land. More specifically, higher resolution promotes grounding line stability in all of these regions for the UB experiments, and particularly promotes instability in Pine Island Glacier, Moscow University Ice Shelf, and Totten Glacier for the IB experiments. Since the IB upper bound ice shelf melt forcing is less extreme than that of the UB forcing for Pine Island and Wilkes Land, the final grounding line position after 100 years falls within regions of locally complex topography (Fig. S5). As a result, the IB experiments emphasize more complicated feedbacks between bedrock geometry and grounding line dynamics. These findings suggest that improved resolution within these basins generally affects model results by refining the bedrock topography, altering stability, and changing the threshold for collapse. Thus, sea level projections are highly dependent on the accuracy of the bedrock topography both beneath the current and any new ice shelf areas.

For the ice sheet model, this is particularly important in areas upstream of the current grounding zone, where our model has a lower horizontal mesh resolution (Fig. S1). For instance, results from our Bedmap1 experiment reveal that the use of a less refined bedrock yields smaller estimates of SLE on a continental scale, suggesting that overall, Bedmap1 lacks the details of the bedrock topography needed to resolve the features in the ice sheet's deep interior. On the 100-year timescale investigated here, features such as bedrock ridges can mitigate ice shelf instability and serve as pinning points, while others, such as troughs and overdeepenings, can promote ice shelf instability (e.g. Fig. S5). Therefore, on a regional scale our results reveal complexity and spatial variability in how bedrock refinement affects grounding line retreat. For example, we find that the largest regional impact of bed topography is in the Amundsen Sea sector under the UB experiment (GP 4, Fig. 6). In this region, the use of Bedmap1 promotes greater regional SLE contribution, yet on a continental scale the Bedmap1 topography results in a reduced total AIS SLE contribution. Such findings highlight the need for continual improvement of high resolution maps of bedrock topography, prioritization of long term measurements of melt rates and ocean temperatures, and the ability for models to capture increased details of the bedrock (particularly in areas near the present and potential future grounding line positions, including the Amundsen Sea sector, the Filchner-Ronne Ice Shelf basin, and coastal Wilkes Land).

While ice shelf melt rates are an important source of uncertainty in the SLE contribution from the AIS, regional results for the IB experiments also highlight the potential for accumulation changes to significantly affect results, particularly in the East AIS. Notably, we find that the Davis Sea sector and Wilkes Land coast (Fig. 4, GP 26 and GP 27, including Moscow University Ice Shelf, Totten Glacier, West Ice Shelf, and Shackleton Ice Shelf) are subject to potential increases in accumulation that can be large enough to mitigate the mass loss resulting from grounding line retreat forced by greater ice shelf melt rates. In these regions, enhanced accumulation can counteract any melt-induced SLE contribution and also widens the simulation uncertainty (Fig. 4). These results suggest that in these coastal areas of the East AIS, the modeling of future ice mass change may be too

uncertain to even determine its sign. As a result, it will be necessary to properly model atmospheric forcing and its response to warming (including refinement of ice sheet and ocean model resolution), particularly within the Wilkes Land (GP 27), Queen Mary Land, and Wilhelm II Land regions (GP 26), in order to reduce uncertainty in future sea level projections.

As the results illustrate, simulations are highly sensitive to oceanic and atmospheric forcing, thus care must be taken to understand how experimental design affects uncertainty. For instance, for the UQ sampling experiments, uncertainty is sensitive to the partitioning methods used (Fig. 2 and Table 3), so our results are a function of our partition boundaries and their chosen bounds. While we consider our adoption of the GP configuration a reasonable approach, this assumption is a model limitation. More specifically, our results suggest that the design of partition configuration requires knowledge about proximal correlations in model forcing associated with changes in climate. For instance, if we underestimate regional correlation by using multiple partitions, this choice may, for accumulation, overestimate the mean of the SLE response distribution and, for ice shelf melt rates, underestimate the SLE uncertainty (or probability of endmember occurrence). Our method also assumes that our sampling variables can be sampled independently, though in actuality there are physical relationships that dictate these variables may vary together in response to warming. For example, climate warming is generally associated with increased ice shelf basal melt rates, increases in accumulation, and decreases in ice viscosity and basal friction. Ideally, our UQ experiments would consider these relationships, however we cannot adequately accommodate them into our present experimental design, especially since the quantification of such relationships is highly uncertain in itself. Therefore, in order to characterize how spatial and variable correlation may affect our results, we conduct additional UQ sensitivity experiments that assume definitive correlations between all variables and within all partitions. Ultimately, these experiments reveal the magnitude of this effect is dependent on the experiment being conducted, particularly the sampling bounds of the forcing variables. For instance, results suggest that for the IB_27GP experiments, continental-wide spatial and variable correlation can decrease the maximum 95% confidence interval by up to 120 mm. Yet, for the UB_GP experiment, correlations could lead to widening of the SLE PDF uncertainty bounds, increasing the maximum 95% confidence interval by up to 172 mm. Such results illustrate the sensitivity and complexity of the system being assessed. Indeed, it is important to note that in the future, the precise quantification of SLR bounds will likely require analysis of a fully coupled atmosphere-ocean-ice model that can physically represent relationships between all ice sheet model forcing.

Additional limitations of our approach include the application of model forcing and control model drift. That is, all model forcing is applied abruptly as a step function at the beginning of each simulation, which will result in a general overestimation of plausible warming scenarios and a larger AIS SLE contribution. Though for the study presented here, we investigate simple perturbations on a control run, this framework can be used to quantify uncertainties in projections using transient forcing. In subsequent studies, users can apply sampling to forward simulations of future projections in order to specifically isolate regional forcing, boundary conditions, and processes that contribute to simulation uncertainty. As noted previously, our simulations also experience model drift, due to general mismatch in model forcing. In many regions, this mismatch outweighs committed changes captured in our instantaneous spin up procedure. We address this issue by subtracting a respective control from all results. Note that because we are simulating a dynamically sensitive system, this approach will not account for committed

changes or associated transient feedbacks. Model sensitivity experiments run with varying mesh resolution and stress balance approximations (e.g. Fig. S2) suggest, however, that such feedbacks are negligible on the timescales addressed here.

Lastly, our simulations do not consider all physical processes, especially those poorly understood but possibly significant drivers of local ice dynamics, including the temporal evolution of basal friction (e.g., Gillet-Chaulet et al., 2016) and ice viscosity (e.g., Khazendar et al., 2015). We address these limitations by considering possible future changes in basal friction and ice viscosity when defining sampling bounds. Note that because we utilize a Budd-type friction law, we likely overestimate changes to the grounding line position (in comparison to a Schoof/Tsai (Schoof, 2005; Tsai et al., 2015) or a Weertman (Weertman, 1957) friction law). Results presented by Brondex et al. (2017) suggest that the Budd-type friction law promotes a more sensitive grounding line migration, with up to three times the magnitude of retreat over a Schoof/Tsai law and thirty times that predicted by a Weertman law for a period of 100 years. While the Budd-type law chosen here advantageously captures a larger range of potential model outcomes, it may also exaggerate endmember SLE contribution by orders of magnitude. Though consideration of different friction laws is outside of the scope of this study, it should be noted that the modeling of basal friction remains a significant uncertainty in ice sheet model projections. Next, because ice fronts remained fixed in time, we do not consider calving or ice shelf breakup events that would reduce ice shelf buttressing. Therefore, we likely underestimate SLE contribution from the AIS deep interior. We address this limitation with the shelf-collapse experiments (Sect. 2.3.2). Overall, we find that the largest effects impact areas where ice fronts are not likely to retreat on the timescales considered here (i.e. the Filchner-Ronne basin, the western Ross basin, and coastal Dronning Maud Land, Fig. 6). Finally, it is important to note that our simulations do not include solid earth feedbacks or associated uncertainties, including glacial isostatic rebound (Barletta et al., 2018) and short-term viscoelastic response (Adhikari et al., 2014), both which may promote grounding line stability on the timescales investigated here. While investigation of the uncertainties associated with a full coupled ice-sheet/solid earth system is outside of the scope of this study, such a future analysis will be imperative for properly constraining uncertainties in projected sea level.

With consideration to these current model limitations, our results suggest that the century-scale potential of SLE contribution from the AIS ranges between 152 mm and 1235 mm of GMSL contribution over the next hundred years, with the more plausible realizations reaching up to 495 mm. Note again that these results quantify only deviations from a control simulation, which means that they do not include any committed SLE contributions that may be captured by our instantaneous spin up procedure ($\sim$ 25 mm over 100 years, as extrapolated from historical estimates of continental mass imbalance, i.e. Shepherd et al. (2018)). Our estimates are comparable to the continental AIS SLE contributions reported by Golledge et al. (2015), who use a coupled ice-sheet/ice-shelf model for the RCP 8.5 scenario (100 mm to 390 mm at year 2100), and they exceed the potential SLE AIS contribution projection by Ritz et al. (2015), who use an ice sheet model forced by the Special Report on Emissions Scenarios AIB scenario (Church et al., 2013) (300 mm at year 2100). Note that results from Golledge et al. (2015) include substantial retreat of Ross, Filchner-Ronne, and Amery Ice Shelves, and much of the reported SLE contribution is due to their local loss of buttressing and the consequential ice dynamic response. Based on our IB instantaneously-collapsed shelf experiment, we bound the 100-year AIS SLE contribution, including ice shelf collapse, to a maximum of 680 mm, exceeding the Golledge et al.

(2015) estimate. Without a coupled ice sheet/ocean system we expect to overestimate interior melt rates, so such a discrepancy is anticipated (Seroussi et al., 2017b).

However, the largest dissimilarities may be due to considerable irreconcilable differences in regional-scale grounding line sensitivity. For instance, the Golledge et al. (2015) simulations show greater grounding line vulnerability in the Recovery/Slessor and the western Ross Ice Shelf / Siple Coast (GP 10) basins, while our results reveal grounding line vulnerability in the western Ronne Ice Shelf, in better agreement with the high resolution regional assessment of Cornford et al. (2015). Since the Ritz et al. (2015) results are based on a different (midrange) warming scenario, it is not surprising that our SLE contribution potential also exceeds their estimates. Yet again, we find that the largest differences are sourced in regional discrepancies. For example, Ritz et al. (2015) report, similar to our findings for our UB experiment, that the Amundsen Sea sector, particularly the Thwaites region (i.e. Fig. S5a), is responsible for almost 50% of their simulated AIS SLE by 2100. In contrast, our more plausible IB experiment results show relatively minor contribution from the Amundsen Sea. Results suggest that this discrepancy may partly be due to a difference in resolution between the two studies (as discussed above, Fig. S4). Comparison with the high resolution regional results from Cornford et al. (2015), who suggest that under a realistic warming scenario the Amundsen Sea would be expected to contribute ∼50 mm SLE contribution by 2100, also supports this hypothesis.

Overall, our results agree with the assessment made by Ritz et al. (2015), that simulating over 1 meter of AIS SLE contribution by 2100 are only robustly achieved under conditions that we consider to be unrealistic (uniform, continental-scale, abrupt changes to oceanic and atmospheric forcing). Indeed, while up to 1.2 meters of AIS GMSL contribution over the next century is achievable, our results suggest that it is not likely.

## 6  Conclusions

In this study, we take advantage of ISSM's highly parallelized, high-resolution UQ framework that allows for regional and local statistical assessment of simulated mass change. We use this framework to sample modeled changes in AIS volume above floatation and to investigate the regions responsible for the largest contribution to uncertainty in estimates of SLE contribution. We also investigate the independent contributions to uncertainties in our simulation from ice shelf melt rates, accumulation, basal friction, and ice viscosity. Results are based on two sets of experiments, each with its own distinct strategy for setting the bounds for the four variables being sampled. In the first case, the bounds represent a generalized warming over the entire ice sheet, while in the second case bounds are set regionally and are informed by literature and model sensitivity studies. Overall, we find that over 1.2 meters of AIS global mean sea level contribution over the next century is achievable, but not likely, even under the a generalized UB experiment, as such a response is only plausible when the model is forced with unrealistically large melt rates and continental ice shelf collapse.

Additionally, for both IB and UB experiments, ice shelf melt rates are responsible for the majority of the uncertainty in the total AIS SLE PDF curves. The second largest contributor is accumulation, followed by basal friction and ice viscosity. For the UB experiments, SLE is dictated largely by the ice shelf melt rate forcing within the Amundsen Sea, accounting for about 47% of the continental AIS mean SLE contribution and about 89% of total AIS SLE uncertainty. In the IB experiment, this region

does not exhibit the same complex behavior that is driven by the generalized UB experiments (i.e. 10 time current melt rates). Instead, under the IB experiment, the Ronne Ice Shelf basin tributaries, specifically Evans and Rutfort Ice Streams, show the greatest potential for SLE contribution and are the largest source of simulation uncertainty. Coastal Wilkes Land, specifically Totten Glacier and Moscow University Ice Shelf, also robustly contribute to simulation uncertainty. In this area, changes in

accumulation are large enough to mitigate mass loss caused by increased melt rates and retreating grounding lines, contributing to uncertainty in the magnitude and sign of regional SLE contribution.

     Recent improvements in the bedrock topography maps in the Ross Ice Shelf basin and in coastal Wilkes Land have increased their estimated potential SLE contributions, while simultaneously decreasing the potential SLE contributions from Pine Island and Thwaites Glaciers. Overall, we estimate that improvements in bedrock topography since Bedmap1 are responsible for a

mean increase up to 0.14 m of AIS SLE contribution over the 100-year period investigated here. Continued refinement of the AIS bedrock topography, additional estimates of melt rates and observations of ocean temperatures, improved ocean and ice sheet model resolution, and better representation of the evolution of ice shelf melt rates within the currently grounded interior are therefore the most important advancements necessary for improving confidence in century scale future projections of Antarctica's contribution to sea level.

*Code and data availability.* RACMO2.1 model output used in this study (Lenaerts et al., 2012) is available upon request from M.R.vandenBroeke@uu.nl. ISSM model output used in this study is available upon request from the ISSM model team (issm@jpl.nasa.gov or http://issm.jpl.nasa.gov/ contactus/) or from schlegel@jpl.nasa.gov. MATLAB code used to analyze model results is also available upon request from schlegel@jpl.nasa.gov. The MC bed topography product is currently under preparation for public release, and more details regarding the product and its release can be obtained from mathieu.morlighem@uci.edu. The ocean model output used in this study is currently under preparation for public release.

Details can be obtained from schodlok@jpl.nasa.gov.

## 7    Appendices

### Appendix A: Determination of informed bounds for ocean forcing

Informed bounds for ocean induced melt rates under ice shelves are derived from a combination of (1) a circumpolar (See Sect. 2.2, Schodlok et al. (2016)) and (2) an idealized MITgcm model setup. The circumpolar model is used to derive realistic

melt rates with optimized turbulent exchange coefficients for each ice-ocean cavity. Since it is expensive to run sensitivity experiments with this setup, we utilize an idealized model (based on De Rydt et al. (2014)) to derive melt rates, with respect to changes in cavity shape (including slope, draft and water column thickness), temperature (-2°C , 0°C , +3°C ), and turbulent exchange coefficients (order of magnitudes $10^{-7}$ to $10^{-3}$ by half power), for a total of ∼660 runs. Each run needed ∼6 months of model integration to reach a steady state.

Simulation results reveal that changes in cavity shape are minor compared to temperature, therefore we average the sensitivity experiment results from the cavity shape runs in order to derive a mean model response (with respect to cavity shape), for the

temperature and turbulent exchange coefficient runs. Then for each temperature simulation we derive a relationship between average ice shelf melt rate and turbulent exchange coefficient. Using the optimized turbulent exchange coefficients from the circumpolar solution and the relationship derived above, we obtain ice shelf melt rates for each set of temperature runs (-2°C , 0°C , +3°C ) within all the ice shelf cavities of Antarctica that are resolved by the circumpolar model set up. As it is the Circumpolar Deep Water (CDW) that provides the heat source to potentially change the melt rates within a particular cavity, we use a linear fit to derive the melt rate for the mean daily maximum temperature of the CDW offshelf of the respective cavity, and calculate the multiplier within each GP with respect to mean melt rates from the optimized solution (See Table S1 for values derived for each GP). The resulting multipliers serve as the maximum values for melt rates in the IB sampling experiment. For GPs that are located inland, melt multipliers are derived from the GPs that are downstream of ice flow, within the same ice drainage basin. The resulting IB melt multipliers represent the potential change in melt rate due to changes in the heat content available offshore, caused by changes in ocean circulation; therefore, they do not represent any future increase in CDW temperatures or consequential impact on melt rates, but instead a maximum potential change for each individual region.

One caveat is that the optimization in some regions are unphysical, despite the fact that mean melt rates agree with those observed (Table S1), leading to unrealistically sensitive melt rates and large upper bound melt rate multipliers. This is particularly the case for ice shelves in GP 12, GP 24, and GP 26. Mismatch in these areas are likely due to model grid spacing, i.e., underrepresentation of the size of the ice shelves, as well as lack of knowledge of the shape of cavity and bathymetry in and outside of the cavity. Note that these sectors contain only minor ice shelves, and do not have a significant impact on results presented in this manuscript. Note also that in some cases, the derived melt multipliers maximum endmember values represent an unphysical regional ice shelf melt rate forcing. For these instances, we cap the melt multiplier values regionally, to the largest value, less than the derived multiplier, that does not result in ice mass removal that exceeds the local mass balance between the divergence of ice flow and surface mass balance. These capped multiplier values are determined through a series of ISSM AIS sensitivity studies varying the magnitude of the melt multiplier in each GP. The GPs affected include: GP 6, GP 12, GP 17, GP 20, GP 24, and GP 26, all areas with large uncertainties in modeled melt rates, as noted above.

For the large ice shelves (e.g., Filchner-Ronne, Ross) and the ice shelves with high sensitivities to melt rates (e.g., Pine Island, Ronne), we are confident that the IB multipliers are well-informed compared to using a generalized melt multiplier (as done for the UB experiments) and that overall, they represent plausible melt rate estimates for potential ocean warming within each region on a century scale.

*Author contributions.* All authors discussed results presented in this manuscript. NS led the design, execution, and analysis of ISSM UQ experiments. HS was responsible for mesh resolution sensitivity runs, determination of IB sampling bounds, and analysis of results. MS was responsible for all ocean model runs and determination of ice shelf basal melt sampling bounds. EL was in charge of the original design and implementation of parallel DAKOTA within ISSM. CB, DL, and MW managed the UQ tasks and were involved in developing and guiding the scientific strategy for this project. MM contributed MC bedrock topography and development of L1L2 stress balance approximation within ISSM. MvdB contributed RACMO2.1 estimates of SMB and associated components.

*Competing interests.* The authors declare that they have no conflict of interest.

*Acknowledgements.* The research was carried out at the Jet Propulsion Laboratory, California Institute of Technology, under a contract with the National Aeronautics and Space Administration. Funding was provided by grants from Jet Propulsion Laboratory Research Technology and Development and from NASA Cryospheric Science and Interdisciplinary Research in Earth Science Programs. We gratefully acknowledge computational resources and support from the NASA Advanced Supercomputing Division. MvdB acknowledges support from the Netherlands earth System Science Centre (NESSC). This work was made possible through model development of the ISSM team. The authors would also like to thank Dr. Amy Braverman for her statistical insight and discussions on model uncertainty. Copyright 2018. All rights reserved.

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

**Figures**

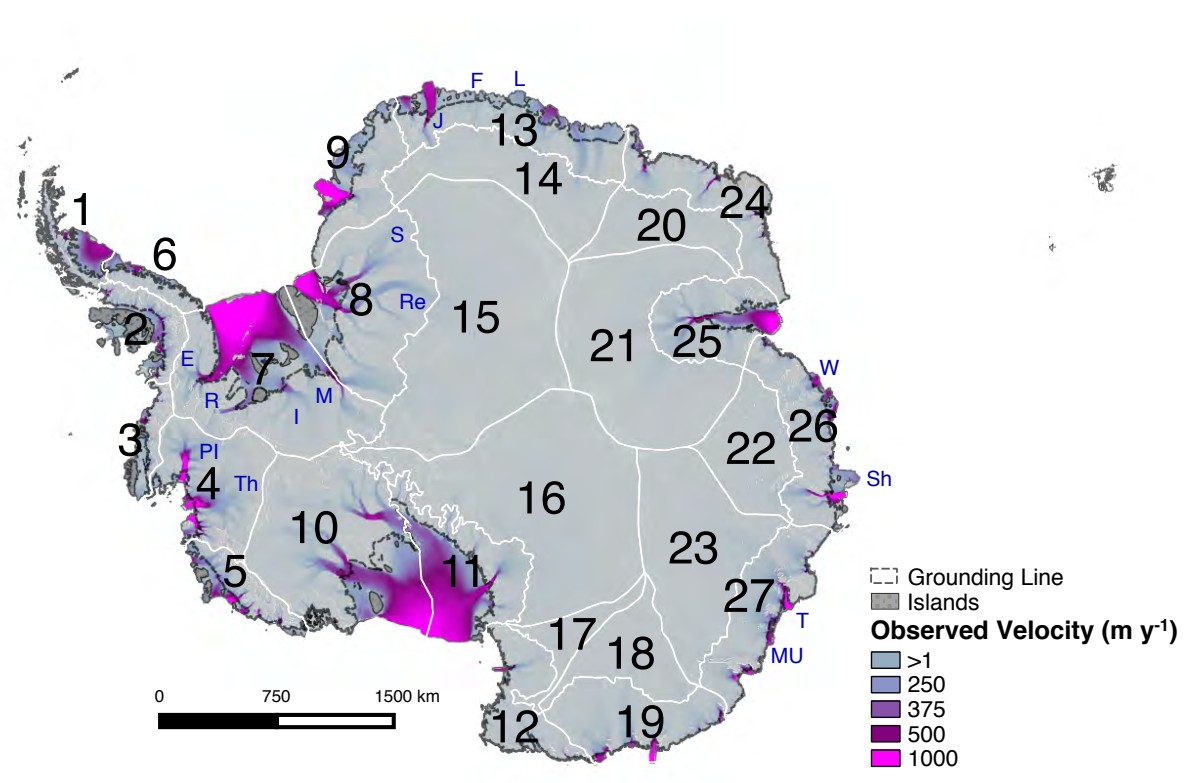

**Figure 1.** Observed surface velocity (m y$^{-1}$) in Antarctica, with Geographic Partitions in white, labeled with their corresponding numerical value (1-27). We indicate the observed grounding line with a dark gray dashed line, and additional features of interest in blue: Th - Thwaites Glacier, PI - Pine Island Glacier, E - Evans Ice Stream, R - Rutford Ice Stream, I - Institute Ice Stream, M - Möller Ice Stream, Re - Recovery Ice Stream, S - Slessor Ice Stream, J - Jutulstraumen Glacier, F - Fimbul Ice Shelf, L - Lazarev Ice Shelf, W - West Ice Shelf, Sh - Shackleton Ice Shelf, T - Totten Glacier, MU - Moscow University Ice Shelf.

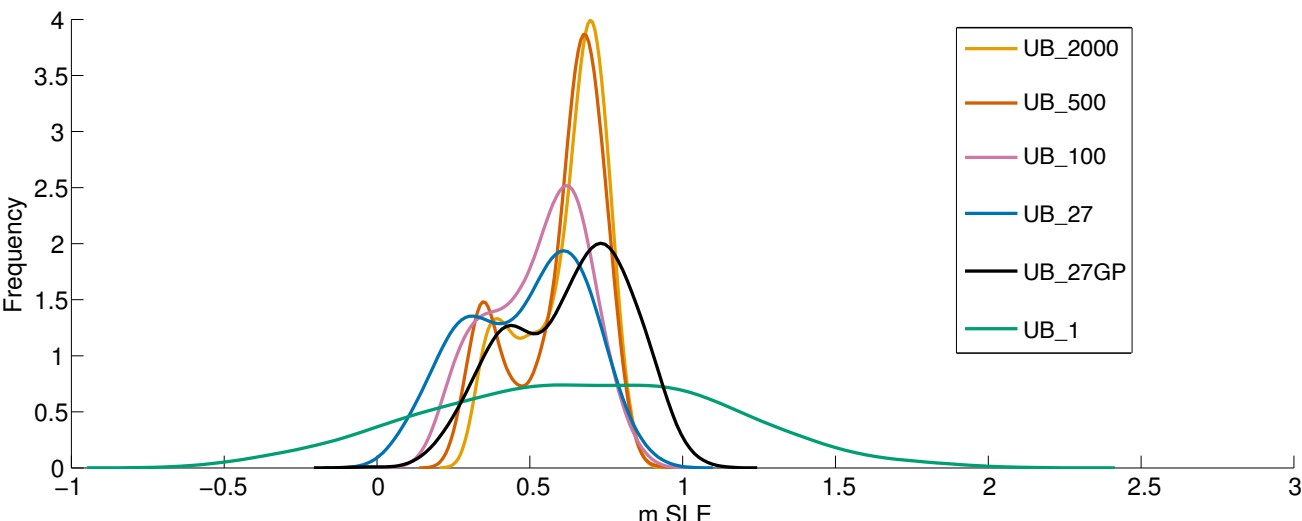

**Figure 2.** PDF of Antarctica SLE (m) contribution after 100 years, for 800 individual simulations sampled with UB and various partition configurations. The distributions result are from sampling experiments where we vary the number and configuration of the sampling regions (partitions) (comparing experiments: UB_2000, UB_500, UB_100, UB_27, UB_1, and UB_27GP). See Fig. S3 for plots of the spatial configuration of partitions for these experiments. See Table 3 for statistics relating to each curve.

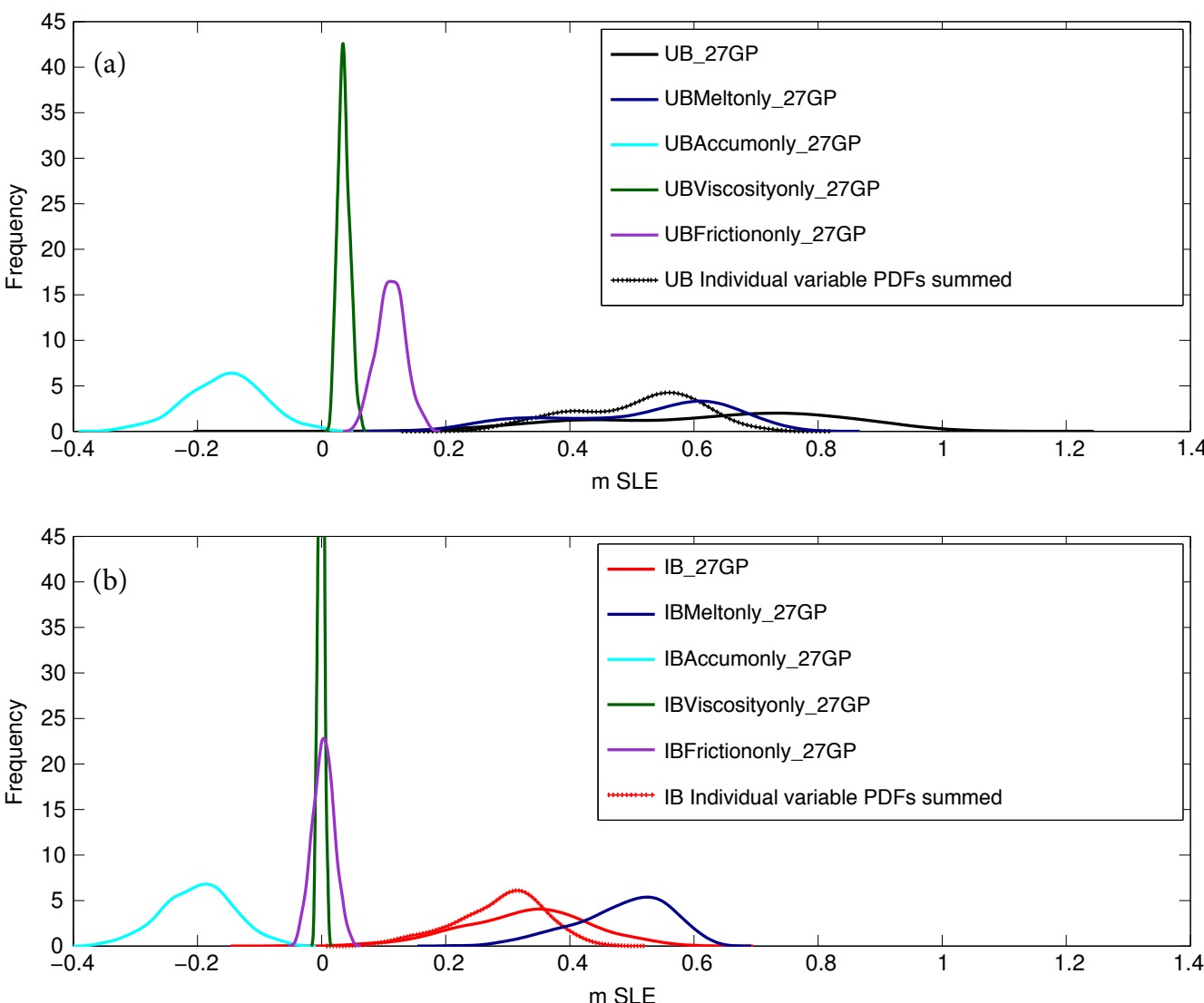

**Figure 3.** PDF of SLE from ensemble runs with 800 simulations using GP partitioning for sampling variables in combination and individually. Both the (a) UB (solid black line) and (b) IB (solid red line) experiments are included. Variables ice shelf basal melt, accumulation, ice viscosity, and basal friction are sampled together randomly (UB_27GP, IB_27GP). Curves of other colors represent the sampling of each variable individually: ice shelf basal melt (UBMeltonly_27GP, IBMeltonly_27GP: dark blue), accumulation (UBAccumonly_27GP, IBAccumonly_27GP: cyan), ice viscosity (UBViscosityonly_27GP, IBViscosityonly_27GP: dark green), and basal friction (UBFrictiononly_27GP, IBFrictiononly_27GP: purple). Hash-marked curves represent the sums of the individual variable curves, for (a) UB (black) and (b) IB (red). See Table 3 for statistics relating to each curve.

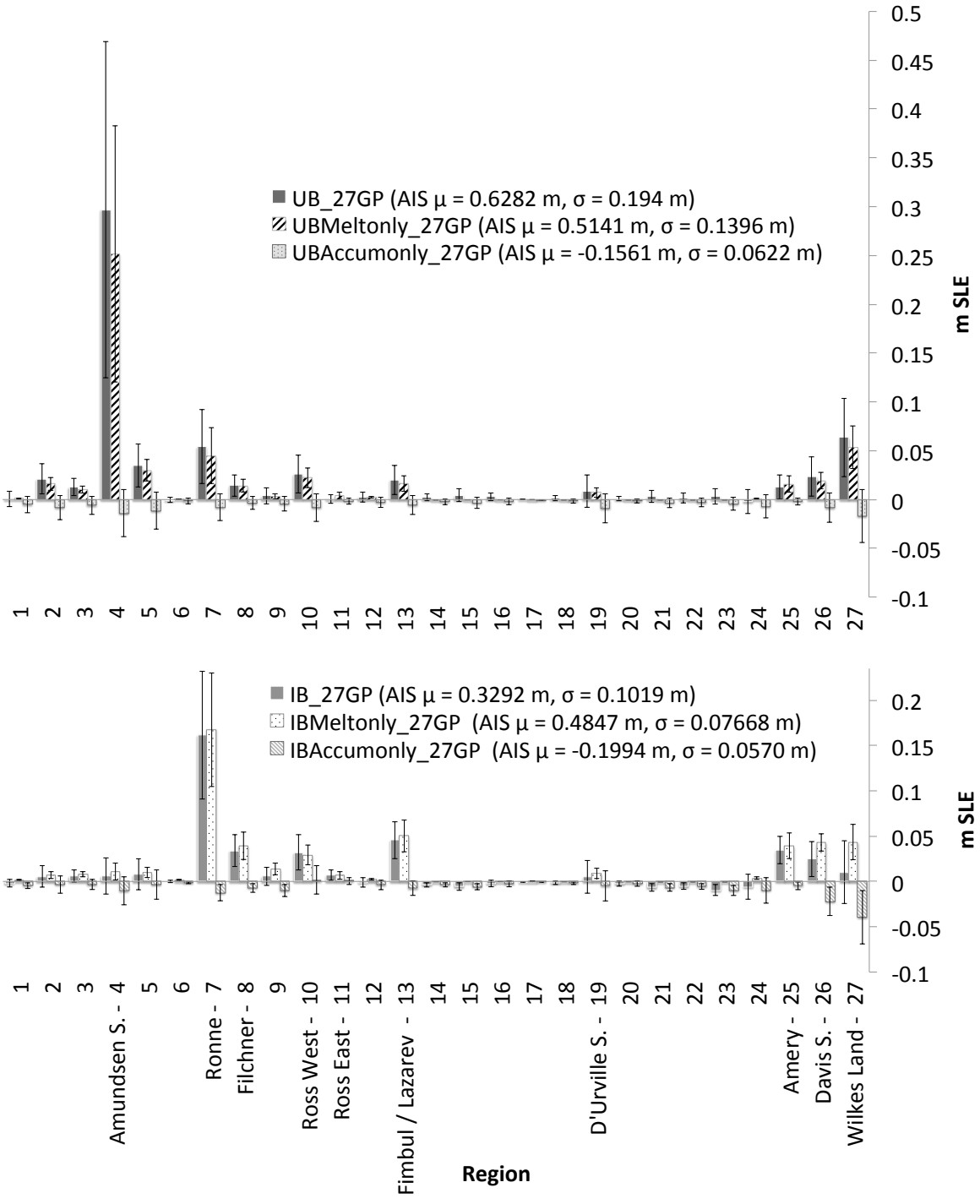

**Figure 4.** Mean SLE (m) and standard deviation bounds for ensemble runs with 800 simulations using GP partitioning, presented regionally, comparing results from the sampling of a combination of variables, of only ice shelf basal melt rates, and of only surface accumulation (top: UB, bottom: IB). Numbers correspond to GP numbers (See Fig. 1). Mean ($\mu$) and standard deviation ($\sigma$) SLE for the total ice sheet are indicated in the legend for each sampling experiment.

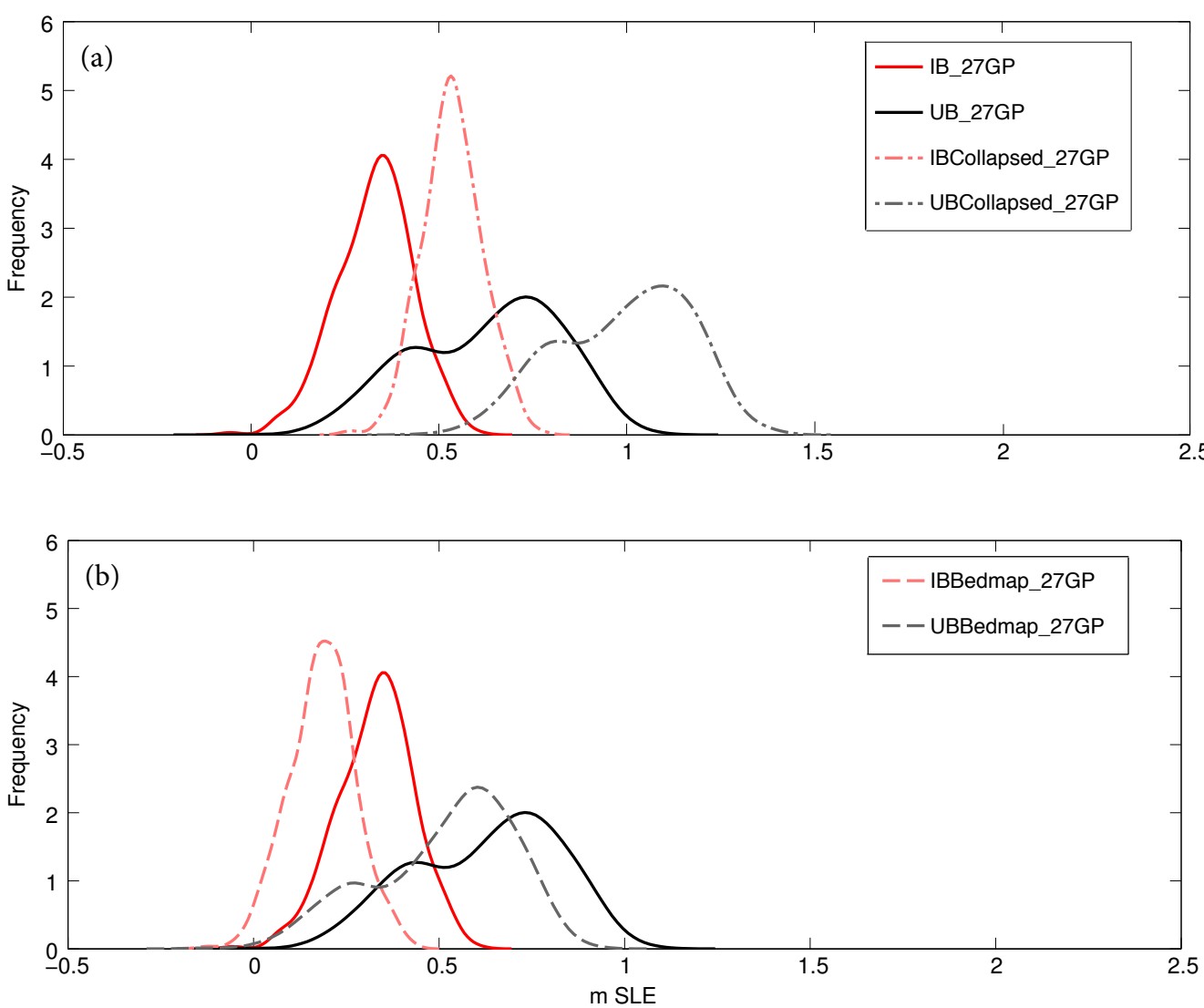

**Figure 5.** PDFs of SLE resulting from ensemble runs with 800 simulations using GP partitioning, comparing the effects of using different boundary forcing. (a) Curves resulting from sampling of UB (UB_27GP, black) and IB (IB_27GP, red) experiments, compared with results from runs forced with collapse of all Antarctic ice shelves at the beginning of the simulation (dash-dot) (UBCollapsed_27GP and IBCollapsed_27GP). (b) Similar plot to (a) but with additional ensemble runs showing the impact of using a bedrock topography (dashed) derived from Bedmap1 (UBBedmap_27GP and IBBedmap_27GP). See Table 3 for statistics relating to each curve.

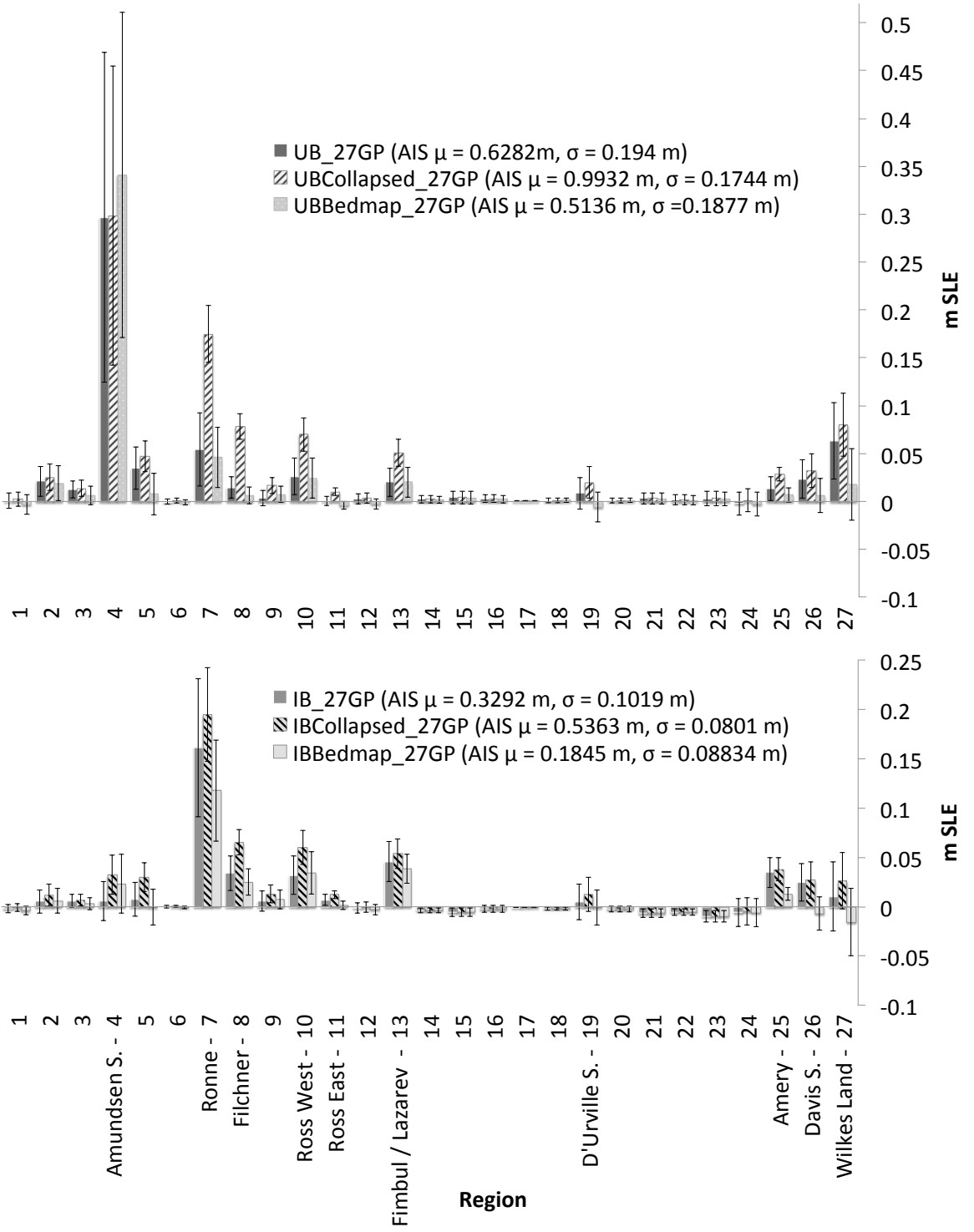

**Figure 6.** Same as Fig. 4 but comparing the combined variable GP experiments, the combined variable GP experiments with ice shelves collapsed, and the combined variable GP Bedmap1 experiments (top: UB, bottom: IB).