# Peer review of "Exploration of Antarctic Ice Sheet 100-year contribution to sea level rise and associated model uncertainties using the ISSM framework"

_The Cryosphere, 2018_

## Referee Comment (RC1) · S. L. Cornford (Referee) · 9 Jul 2018

This paper presents a thorough ensemble exploration of the sensitivity of a state-ofthe art Antarctic ice flow model (ISSM) to uncertainty in near future perturbations. To my mind, it addresses the shortcomings of most previous attempts by (1) considering the whole continent (2) doing so at (I think) adequate mesh resolution (3) including a well thought out and well described experimental design. It ought to be a benchmark against which the community measure future efforts for some time. Given that, I don't really have any substantial criticisms of this paper, just a few minor comments to make

Abstract, L11 : it seems odd to say that 'grounding line retreat... is \*driven by errors\* in

bedrock retreat'. Rephrase?

Abstract L13 : 'endmember' -> 'the most extreme'?. endmember seems a little 'jargon' for the abstract.

p3, L16 '1km resolution at the coast' - where is the coast - the calving front? Or the grounding line?. On this note, is the region over which the GL moves generally covered by  $\sim$ 1km mesh spacing? Could we have a figure?

P4, L6 : L1L2 costs 10X SSA - this is a surprise (my own L1L2 is about 1.2X costlier than SSA), can you say why? There are two things in L1L2 that might slow you down (a) a more costly viscosity calculation (b) shorter timesteps due to the diffusion-like compenent of the velocity, but these a modest for me. Or is there something about your SSA that makes it very fast (e.g, how many Newton iterations are carried out per time-step). Possibly this is described in the cited papers, but a quick summary here would be helpful.

P5, L11. Do you mean that the presented results are 'simulation minus control' or something else?

P5,L14 : You would see the vast majority of the change if the ice shelf thinned to 1m, even if the front remained the same. I think here you are adding the result of immediate collapse, rather than saying much about gradual future calving front retreat.

P7,L30 'plus an epsilon of  $0.01\%' \rightarrow$  'plus 0.01%'? I think know what you mean by an epsilon, but it seems redundant, here (0.01% is a small change compared to -40%).

P9, L7. If the melt rates are too great in some places and too small elsewhere, why is it reasonable? Not saying it isn't, just that you should either say why, or not say it at all.

P10: First paragraphs of result. A bit of rewriting is needed here. The second paragraph in particular is describing methods, rather then (say) outlining the results/structure of the results section.

TCD
P13:first paragraph. I think here you are saying that as the number of partitions grow, you are seeing some result other than just the number of neutral outcomes grow quickly. OK, your mean is decaying, not just the spread, but I can't see from what is written what (substantive) evidence you have for that position.

P13:L30 – looking at the figure, I thinking you are concluding that the bimodal shape comes from the melt rate UQ because only that parameter gives a bimodal distribution when considered in isolation. Should you say that outright in the text?

P17, L27: Durand, Gladstone, and others (inclusing Seroussi, no?) have all indicated a resolution very much finer than 8km is required around the grounding line (I find  $\sim$ 1km, and sometimes better, is needed for Antartica, even with SEP treatments). But I thought (from p3, L16) that you might be at least close to this?

P19, L10: Is control mode drift necesarrility undesirable? The model parameters are optimized to given the present day velocity / geometry, so ought to agree with the present day observed thinnig rate, which is not zero – there is comitted change to think about. Given the general nonlinearity of, say Thwaites glacier retreat, I think is is better to have the correct drift, than none.

TCD

---

## Referee Comment (RC2) · Anonymous Referee #2 · 18 Jul 2018

**Comment on 'Exploration of Antarctic Ice Sheet 100-year contribution to sea level rise and associated model uncertainties using the ISSM framework'**

presented on 1st of June 2018 by Schlegel et al.

In this manuscript, an uncertainty assessment of the 100-year contribution to sea-level rise from the Antarctic Ice Sheet is presented. Future sea level rise is a highly important topic of great interest for the scientific community and of large societal importance. The study by Schlegel et al. uses a statistical uncertainty quantification method that has been applied previously for regional setups (e.g., Larour et al., 2012a,b; Larour and Schlegel, 2016; Schlegel et al., 2013, 2015) and is now applied to an Antarctic-wide setup. The method is based on a number of forward model runs, which are performed with a resolution of  $\leq 8$  km in ice streams and ice shelves, solving the 'SSA' using the ISSM ice-sheet model.

The authors test for the effects of uncertainties related to surface accumulation, sub-shelf melting, ice viscosity as well as basal friction on mass loss from the ice sheet. Schlegel et al. specify 'perturbation bounds' (uniform or individually for each drainage basin) for each of these variables. Using Latin Hypercube Sampling, a statistically representative number of perturbations for each variable and each region ('partition') of the model domain is determined. The perturbations are applied as step-forcings, uniformly over the corresponding partition. Based on these experiments, a statistical distribution of sea-level relevant ice volume loss after 100 years of model time is derived and analysed.

In additional experiments, Schlegel et al. test for the role of the bed geometry measurements. Their findings highlight that the details of bed geometry are decisive for grounding line movement that determines mass loss. In further experiments, they also test for the effect of an instantaneous ice-shelf collapse (however ice shelves form in regions that unground during grounding line retreat). The manuscript is well written and provides much detail on the model experiments and the results. The methodology presented is innovative and provides valuable insights into uncertainties of Antarctic mass loss. I have only a few major comments on the manuscript:

**Major comments:**

• Sub-shelf melt rates With your methodology, you find that sub-shelf melt rates are dominating the mass loss uncertainty. It would hence be great to have more information on the melt rate field you use, e.g., a figure showing a spatial map of the reference sub-shelf melt rates. How large is the spread of melt rates - within ice-shelves and in-between ice shelves? Are there regions with accretion? How do the melt rates compare to observations? How large are the melt rates near the grounding lines, which are then also applied in newly ungrounded regions? And how large are the maximum CDW values that you use to constrain the IBs for melting in the individual partitions?

**• UQ and partitions**

Please add more details on the UQ method, its underlying assumptions, benefits and limitations, the way it is applied here and on the choice of partitions made.

In previous applications of the uncertainty quantification (UQ) technique, a high number of partitions has been used ( $\geq 500$  for regional setups in Larour et al., 2012a,b; Larour and Schlegel, 2016; Schlegel et al., 2013, 2015). Since uniform perturbations are applied within each partition, increasing the number of partitions yields convergence of the resulting uncertainty distribution as long as the errors/uncertainties sampled within the individual partitions are independent, as far as I understand. But the uncertainties under future warming scenarios, e.g., of surface accumulation changes, are not spatially uncorrelated, and this is why you find a large dependency of you results on partitioning (Sect. 4.1). How does the application of the UQ in this study hence differ from the previous, regional applications (where a large number of partitions was used)? Please discuss this in the manuscript.

Since the IB experiments based on partitions derived from drainage basins are expected to yield more realistic estimates, it would be highly interesting to understand the effect of partitioning in these experiments: if you do not split drainage basins in high surface elevation regions and low surface elevation regions, how does this affect your results (e.g. in the IBMeltOnly and IBAccumOnly experiments)? And how are the results affected by combining the partitions relating to the same ice shelf or region (e.g., 7 and 8 for FRIS, 10 and 11 for Ross or 1, 2 and 6 for the Antarctic Peninsula)?

**Minor comments**

page 1, line 13: Maybe omit 'instantaneous' here, since you do not test for ice-shelf collapse during the model run and your statement could be read to say 'it is not possible to achieve > 1.2 meters with a collapse during the run'.

- page 1, line 14 and line 17, page 2, line 25,...: Using the term 'scenario' could be misleading. I did understand this first in the sense that you are making projections of Antarctica's future sea-level contribution, while you are, in fact, assessing uncertainties by using idealized, step-forcing experiments.
- page 1, line 15: Maybe combine the sentences to make clear that this estimate is linked to the UB experiment?
- page 2, line 6: Maybe add that 30cm was the upper bound in (Ritz et al., 2015) and that > 1m was found for RCP8.5 by (DeConto and Pollard, 2016).
- page 3, line 18: What do you mean by 'coast' the grounding line or the ice front? How large is the resolution around the grounding line (see also page 17, line 21). How is the resolution in areas where the grounding line retreats during the model run? Do you refine your mesh during grounding line retreat?
- page 3, line 5: What model parameters do you use, e.g., related to Glen's flow law or the Budd-type friction law?
- page 3, line 29: Why do you invert for ice viscosity in the floating areas instead of applying the same procedure as for the grounded areas (assuming thermal steady state)? Does this introduce jumps in ice viscosity across the grounding line?
- page 4, line 21: Are the exchange parameters optimized per shelf or per region or continentwide?
- page 4, line 32: Please clarify the 'large melt multipliers in the ice sheet interior'.
- page 5, line 5: Are there regions where the grounding line advances? How do you specify basal friction in these areas?
- page 5, line 19: This is a very interesting and important assessment. I wonder if (just for curiosity, if you did not do this, I do not ask you do to so): did you also compare to the original Bedmap2?
- Section 2.4: Please give more details on the underlying assumptions, capabilities and limitations of the UQ method.
- Section 2.4: In this section you relate to 'errors' in the model boundary conditions (e.g., lines 21, 24, 28). This might be confusing the reader since you quantify the effect of uncertainties in the variables rather than errors.
- page 5, line 23: Reading your manuscript, I did understand that you use Latin-Hypercube sampling and not Monte-Carlo sampling (also stated in the abstract)?

- page 5, line 27: You state that you assume a uniform distribution of uncertainties to sample the tested variables (sub-shelf melt rates, surface accumulation, ice viscosity and basal friction). This sounds reasonable given that we do not know how future greenhouse gas emissions evolve or which RCP scenario is most likely and how boundary conditions change in the future. Could you elaborate more on this in the manuscript: why did you chose this type of distribution and how does this affect your results, e.g., in comparison to using normal distributions?
- page 5, line 31: Is there a specific reason that partitions should have equal areas? And how does it affect the IB-results that the GP do not have equal areas?
- page 6, line 3: Please explain.
- page 6, line 11: Could you solve the problem of how to define reasonable partitions by specifying forcing scenarios for the variables (where possible, e.g. for the surface accumulation based on RCP scenario projections) and by using the UQ method to quantify uncertainties imposed on these?
- page 7, line 24: Can you give an estimate to what amount of surface warming this extreme scenario corresponds?
- page 9, line 5: How large do melt rates of FRIS and in the Amundsen sea get?
- page 9, line 15-20: You could state here, that these bounds also encompass all RCP projections (if this is correct). This would underline that you are not only interested in the strongest emission scenario, but that you quantify uncertainties encompassing all scenarios.
- page 10, line 17: Is it valid to assume that variables can be sampled independently from each other and that partitions can be sampled independently? Please add a discussion.
- page 13, line 7: mass 'gain' instead of 'loss' .
- page 13, line 31: The bi-modality is very interesting do you have further hints on how this arises? Is it linked to instability within a specific basin or ice stream?
- page 14, line 26: How large are melt rates around the grounding line in your control run?
- page 14, line 29: How does the mean SLE from the Amundsen Sea Region for the IB experiment compare to observed rates of sea-level rise (e.g. Shepherd et al., 2018)?
- page 14, line 35: The intrusion of relatively warm water into the Filchner-Ronne ice shelf cavity has been found possible within the second half of the 21st century for the RCP8.5 scenario (Hellmer et al., 2012). This would be much later than in the step forcing experiments applied here how is this reflected within your uncertainty quantification method?

- page 15, line 3: Could you add a figure with an example for grounding line migration?
- page 15, line 16: Maybe reformulate this to 'subsequent reduction in buttressing'. The current formulation might cause misunderstanding, since you account for the reduction of buttressing through changes in sub-shelf melting.
- page 17, line 8: Please explain what you mean with 'well-behaved' and 'normally'? Does this relate to the next sentence about the bimodal structure?
- page 17, line 15: E.g., shorten to '... bedrock geometry and ice-shelf buttressing, which determine grounding line stability.'
- page 18, line 10: Why does your reasoning presented here for the UB experiments (lower resolution provides less bed geometry details and this yields more SLE) not apply to the IB experiments?
- page 19, line 14: Another process missing might be isostatic rebound, as recent evidence suggests (Barletta et al., 2018).
- Appendix A, page 21, line 31: This setup was created as an idealized version of the cavity underneath Pine Island Glacier ice shelf. Did you test also for cavity sizes and shapes comparable to FRIS?
- Appendix A, page 22, line 5: 'average ice shelf melt rate'?
- Appendix A, page 22, line 23: Please clarify 'that does not artifically restrict ice flow'.
- Table 2: Please specify 'mean max melting rate'.
- Figure S1: The upper panel looks squeezed in y-direction. The line colors to indicate the sectors in the lower panels are hard to see.
- Figure S5: It is hard to see the grounding line positions, especially the initial position.

**Technical issues**

- page 4, line 1: remove 'are' before 'described'.
- page 10, line 7: 'actual uncertainty'
- page 17, line 10: There is a verb missing in '.., as evidence by..'
- page 20, line 12: This sentence is doubled.
- page 21, line 29: 'setup'

**References**

- Barletta, V. R., Bevis, M., Smith, B. E., Wilson, T., Brown, A., Bordoni, A., Willis, M., Khan, S. A., Rovira-Navarro, M., Dalziel, I., et al. (2018). Observed rapid bedrock uplift in amundsen sea embayment promotes ice-sheet stability. *Science*, 360(6395):1335–1339.
- DeConto, R. M. and Pollard, D. (2016). Contribution of antarctica to past and future sea-level rise. *Nature*, 531(7596):591–597.
- Hellmer, H. H., Kauker, F., Timmermann, R., Determann, J., and Rae, J. (2012). Twenty-first-century warming of a large antarctic ice-shelf cavity by a redirected coastal current. *Nature*, 485(7397):225.
- Larour, E., Morlighem, M., Seroussi, H., Schiermeier, J., and Rignot, E. (2012a). Ice flow sensitivity to geothermal heat flux of pine island glacier, antarctica. *Journal of Geophysical Research: Earth Surface*, 117(F4).
- Larour, E., Schiermeier, J., Rignot, E., Seroussi, H., Morlighem, M., and Paden, J. (2012b). Sensitivity analysis of pine island glacier ice flow using issm and dakota. *Journal of Geophysical Research: Earth* Surface, 117(F2).
- Larour, E. and Schlegel, N. (2016). On issm and leveraging the cloud towards faster quantification of the uncertainty in ice-sheet mass balance projections. *Computers & Geosciences*, 96:193–201.
- Ritz, C., Edwards, T. L., Durand, G., Payne, A. J., Peyaud, V., and Hindmarsh, R. C. (2015). Potential sea-level rise from antarctic ice-sheet instability constrained by observations. *Nature*, 528(7580):115– 118.
- Schlegel, N., Larour, E., Seroussi, H., Morlighem, M., and Box, J. (2013). Decadal-scale sensitivity of northeast greenland ice flow to errors in surface mass balance using issm. *Journal of Geophysical Research: Earth Surface*, 118(2):667–680.
- Schlegel, N.-J., Larour, E., Seroussi, H., Morlighem, M., and Box, J. (2015). Ice discharge uncertainties in northeast greenland from boundary conditions and climate forcing of an ice flow model. *Journal of Geophysical Research: Earth Surface*, 120(1):29–54.
- Shepherd, A., Ivins, E., Rignot, E., Smith, B., van den Broeke, M., Velicogna, I., Whitehouse, P., Briggs, K., Joughin, I., Krinner, G., et al. (2018). Mass balance of the antarctic ice sheet from 1992 to 2017. *Nature*, 556:pages219–222.

---

## Author Comment (AC1) · 17 Aug 2018

We would like to thank the referees for bringing up thoughtful and insightful discussion points, and especially for taking their time to assess our manuscript. Your suggestions and comments are most helpful in improving our overall presentation. Below, we address your questions and concerns. For those comments that required updates to text and/or figures, we enclose a draft with corresponding changes in red.

[Figure]

This paper presents a thorough ensemble exploration of the sensitivity of a state-of-the-art Antarctic ice flow model (ISSM) to uncertainty in near future perturbations. To my mind, it addresses the shortcomings of most previous attempts by (1) considering the whole continent (2) doing so at (I think) adequate mesh resolution (3) including a well thought out and well described experimental design. It ought to be a benchmark against which the community measure future efforts for some time. Given that, I don't really have any substantial criticisms of this paper, just a few minor comments to make

Abstract, L11 : it seems odd to say that 'grounding line retreat... is *driven by errors* in bedrock retreat'. Rephrase?

This has been updated to read "variations in bedrock topography".

Abstract L13 : 'endmember' => 'the most extreme'?. endmember seems a little 'jargon' for the abstract.

We have changed "endmember" in the abstract to "our most extreme".

p3, L16 '1km resolution at the coast' - where is the coast - the calving front? Or the grounding line?. On this note, is the region over which the GL moves generally covered by ~1km mesh spacing? Could we have a figure?

We have updated this sentence to specify that the 1 km resolution is at "... the domain boundary ... with a resolution of 8 km or finer within the boundary of all initial ice shelves". In addition, in the discussion, we have added statements to clarify our caveat that, in the interior, where the most extreme simulations migrate, our resolution is lower than 8 km (See Fig. S1 and Fig. S5), specifically on Page 17, line 27 and Page 18, line 17. In these sentences, we have also added additional references to the supplemental figures of the mesh and the extreme grounding line locations. Unfortunately, for the continental model, we were not able to run with a 1 km resolution over the entire area of extreme grounding line migration. This is something we will try to improve in the future, since it will be increasingly important. The increasing computational capacity of our cluster will help with this as well!

P4, L6 : L1L2 costs 10X SSA - this is a surprise (my own L1L2 is about 1.2X costlier than SSA), can you say why? There are two things in L1L2 that might slow you down (a) a more costly viscosity calculation (b) shorter timesteps due to the diffusion-like component of the velocity, but these a modest for me. Or is there something about your SSA that makes it very fast (e.g, how many Newton iterations are carried out per time-step) . Possibly this is described in the cited papers, but a quick summary here would be helpful.

Thank you for this question. The first thing that slows us down with the L1L2 is that we are carrying 2x the number vertices. Secondly, we have to take shorter timesteps for L1L2. Thirdly, the SSA indeed converges much quicker. This last one is not modest, specifically because UQ requires that we tighten our stress balance relative convergence tolerance - down to at least the order of $10^{-5}$. Doing so ensures that any small divergence in the sampling results is real and not indicative of noise/numerical drift. We have determined the required tolerance threshold experimentally. While this is more important when we are conducting sensitivity studies (where we forcing the model with very small deltas), we generally follow this practice for all UQ analyses in

order to reduce noise. We have added a couple sentences here, noting these reasons for the difference in efficiency.

P5, L11. Do you mean that the presented results are 'simulation minus control' or something else?

Yes. We have updated the sentence to clarify that we present each result "as its simulated value minus that of its respective control simulation".

P5,L14 : You would see the vast majority of the change if the ice shelf thinned to 1m, even if the front remained the same. I think here you are adding the result of immediate collapse, rather than saying much about gradual future calving front retreat.

We have revised this statement to say that we are attempting to quantify "loss of buttressing due to ice shelf collapse".

P7,L30 'plus an epsilon of 0.01%' => 'plus 0.01%'? I think know what you mean by an epsilon, but it seems redundant, here (0.01% is a small change compared to -40%).

Where used, we have removed the word epsilon and just refer to the additive change of 0.01% as suggested.

P9, L7. If the melt rates are too great in some places and too small elsewhere, why is it reasonable? Not saying it isn't, just that you should either say why, or not say it at all.

This is a good point. We have revised this sentence to point out the difficulty in deciding on a single value for the entire ice sheet, now reading "This dichotomy illustrates

the difficulty in designing an appropriate ice shelf melt rate perturbation experiment for Antarctica, as typically, a single offset or multiplier value is chosen to represent a generalized warming scenario". We have also moved the approximation of maximum warming for Pine Island Glacier from the results section to this paragraph in order to inform the reader in the methods, of the extremity of a 10 times melt rate in WAIS.

P10: First paragraphs of result. A bit of rewriting is needed here. The second paragraph in particular is describing methods, rather then (say) outlining the results/structure of the results section.

We have removed the second paragraph from the results section, and have moved the statement about the efficiency of LHS sampling to the Uncertainty Quantification section in the methods. Thank you for this suggestion.

P13:first paragraph. I think here you are saying that as the number of partitions grow, you are seeing some result other than just the number of neutral outcomes grow quickly. OK, your mean is decaying, not just the spread, but I can't see from what is written what (substantive) evidence you have for that position.

This paragraph is meant only to discuss the variation in uncertainty (standard deviation between the curves). It was very confusing, because we had included a statement about a shift in the curves. We believe this is the statement you are addressing, and it has been removed. The paragraph has also been reworked to be more clear for the reader. Note that we discuss the shift in the mean of the curves and the relation to accumulation forcing two paragraphs below, so the removed statement was indeed misplaced. In this paragraph, we are noting only that with less partitions more spatial correlation are preserved, and therefore the model can dynamically respond in more extreme ways. Because of this, we find more endmember realizations with less number of partitions.

P13:L30 - looking at the figure, I thinking you are concluding that the bimodal shape comes from the melt rate UQ because only that parameter gives a bimodal distribution when considered in isolation. Should you say that outright in the text?

We have rewritten these sentences to say that the melt rate is the only variable with a bimodal distribution, and therefore, we are concluding that it is the source of this complex behavior in the all-variable distribution curve.

P17, L27: Durand, Gladstone, and others (inclusing Seroussi, no?) have all indicated a resolution very much finer than 8km is required around the grounding line (I find ~1km, and sometimes better, is needed for Antartica, even with SEP treatments). But I thought (from p3, L16) that you might be at least close to this?

Near areas of high shear, we do approach 1 km, however, we do not guarantee that we reach this resolution around the grounding lines. This was simply not possible with our computational constraints, and therefore, we discuss that caveat here in the discussion. Your point, however is well taken, especially in light of our results. As discussed here, however, comparison with runs using a higher resolution mesh did not substantially change our results for the most extreme cases, particularly in WAIS which is our most sensitive region. Ideally, in the future, we will be able to run with increased resolution near the grounding line and upstream of the grounding line. As discussed above, we have rephrased the methods description of the mesh to make this more clear.

P19, L10: Is control mode drift necesarrility undesirable? The model parameters are optimized to given the present day velocity / geometry, so ought to agree with the present day observed thinnig rate, which is not zero - there is comitted change to think about. Given the general nonlinearity of, say Thwaites glacier retreat, I think is is better to have the correct drift, than none.

You make a good point here. However, we have decided to ignore committed change, because the drift in our model is much larger than the committed change. During the control run, we find that thickening in the interior outweighs committed change in many places (WAIS is probably the exception), and therefore, we have decided to remove it. As discussed in the manuscript, we ran a number of comparison sensitivity experiments with various resolutions, particularly in the Thwaites region. We found that the drift was most sensitive to model resolution, while response to the forcing was relatively consistent. When subtracting out the control, results converged in an acceptable way (and after a number of iterations, this is partly how we compromised on a final mesh that would be computationally affordable and representative of a realistic change (*Seroussi et al.*, 2017)). We have added a sentence to the discussion here to state this point more clearly.

**2  Anonymous Referee 2**

In this manuscript, an uncertainty assessment of the 100-year contribution to sea-level rise from the Antarctic Ice Sheet is presented. Future sea level rise is a highly important topic of great interest for the scientific community and of large societal importance. The study by Schlegel et al. uses a statistical uncertainty quantification method that has been applied previously for regional setups (e.g., *Larour et al.* (2012b,a); *Larour and Schlegel* (2016); *Schlegel et al.* (2013, 2015)) and is now applied to an Antarctic-wide setup. The method is based on a number of forward model runs, which are performed with a resolution of $<= 8$ km in ice streams and ice shelves, solving the 'SSA' using the ISSM ice-sheet model.

[Figure]

The authors test for the effects of uncertainties related to surface accumulation, sub-shelf melting, ice viscosity as well as basal friction on mass loss from the ice sheet. Schlegel et al. specify 'perturbation bounds' (uniform or individually for each drainage basin) for each of these variables. Using Latin Hypercube Sampling, a statistically representative number of perturbations for each variable and each region ('partition') of the model domain is determined. The perturbations are applied as step-forcings, uniformly over the corresponding partition. Based on these experiments, a statistical distribution of sea-level relevant ice volume loss after 100 years of model time is derived and analysed.

In additional experiments, Schlegel et al. test for the role of the bed geometry mea-surements. Their findings highlight that the details of bed geometry are decisive for grounding line movement that determines mass loss. In further experiments, they also test for the effect of an instantaneous ice-shelf collapse (however ice shelves form in regions that unground during grounding line retreat).

The manuscript is well written and provides much detail on the model experiments and the results. The methodology presented is innovative and provides valuable insights into uncertainties of Antarctic mass loss. I have only a few major comments on the manuscript:

**Major comments:**

- Sub-shelf melt rates // With your methodology, you find that sub-shelf melt rates are dominating the mass loss uncertainty. It would hence be great to have more information on the melt rate field you use, e.g., a figure showing a spatial map of the reference sub-shelf melt rates. How large is the spread of melt rates - within ice-shelves and in-between ice shelves? Are there regions with accretion? How do the melt rates compare to observations? How large are the melt rates near the grounding lines, which are then also applied in newly ungrounded regions? And how large are the maximum CDW values that you use to constrain the IBs for melting in the individual partitions?

We have added a table to the supplement that is meant to answer most of the questions you pose here. The table includes mean and standard deviation ice shelf melt rate values for every marine-based partition and for the present-day grounding line areas for these partitions. The table also includes regional mean estimates of ice shelf melt rates from *Rignot et al.* (2013) for comparison. Finally, we include the mean maximum daily cavity temperature and mean maximum daily boundary temperature (or CDW temperature) for each marine-based partition. We believe the inclusion of this table will be very helpful for the reader to better characterize our ocean forcing. Thank you for the suggestion!

We are also currently preparing a manuscript with a detailed analysis of the circum-polar model simulation used to force our control simulation, therefore we do not plan to publish a spatial map of the mean melt rates in this manuscript. However, we are happy to provide a spatial map of the results by personal communication (See Code and data availability section). In addition, we refer the reader to *Schodlok et al.* (2016), which shows spatial results from the same optimization procedure, but with a lower
resolution grid spacing. The results used for this study generally compare well with those presented in *Schodlok et al.* (2016). However, we decided to use the new 9 km grid product because it better resolves melt rates in smaller ice shelves (a significant source of uncertainty, as discussed in the manuscript), and has an improved spatial pattern in some major ice shelves including melt (e.g. in Ross) and of refreeze (e.g. in Filchner Ronne, Amery, and Larsen C). This is evidenced by the good agreement against observations regionally (new Table S1), and the fact that our control simulation grounding line positions match well with observed (as noted in the text).

- UQ and partitions // Please add more details on the UQ method, its underlying assumptions, benefits and limitations, the way it is applied here and on the choice of partitions made. In previous applications of the uncertainty quantification (UQ) technique, a high number of partitions has been used ($>= 500$ for regional setups in *Larour et al.* (2012b,a); *Larour and Schlegel* (2016); *Schlegel et al.* (2013, 2015)). Since uniform perturbations are applied within each partition, increasing the number of partitions yields convergence of the resulting uncertainty distribution as long as the errors/uncertainties sampled within the individual partitions are independent, as far as I understand. But the uncertainties under future warming scenarios, e.g., of surface accumulation changes, are not spatially uncorrelated, and this is why you find a large dependency of you results on partitioning (Sect. 4.1). How does the application of the UQ in this study hence differ from the previous, regional applications (where a large number of partitions was used)? Please discuss this in the manuscript.

We have expanded the uncertainty quantification methods section to discuss a number of challenges/limitations with our methods as well as the difference between using: a large number of small, random partitions vs. the user-defined partition approach (larger partitions informed scientifically). We find that the use of many small partitions is more appropriate for sampling forcing errors (i.e. when an error map is available), when the user wants to propagate these errors through a simulation and better constrain

output uncertainties due to the input errors themselves. Climatological uncertainties are more complex, in that they are spatially correlated and tend to be associated with a much larger spread (uncertainty). The use of many small partitions, in our opinion, artificially restricts the bounds of our sea level diagnostic (as evidenced by the results we present using different numbers of random partitions). These results inspired us to create geographical partitions, but we also included in the manuscript results from the various partition configurations in order to emphasize the impact of experimental design. As discussed in the manuscript, we feel that geographical partitions are more appropriate for our purposes, though our partition choice is surely debatable (and, we hope, improved during future studies). We have now expanded upon this point in the uncertainty quantification section.

Also, we now elaborate on how this study is different from those conducted in the past using ISSM-DAKOTA sampling, and note that in the future, users will need to consider their scientific questions when designing experiments and choosing partitions. In the discussion, we have added some statements expanding upon the limitation of using a step function, as done here. While currently, we do not have temporally-based sampling, it is possible to perturb a transiently forced forward simulation. While out of the scope of this study, we hope to pursue such an analysis in the future.

Since the IB experiments based on partitions derived from drainage basins are expected to yield more realistic estimates, it would be highly interesting to understand the effect of partitioning in these experiments: if you do not split drainage basins in high surface elevation regions and low surface elevation regions, how does this affect your results (e.g. in the IBMeltOnly and IBAccumOnly experiments)? And how are the results affected by combining the partitions relating to the same ice shelf or region (e.g., 7 and 8 for FRIS, 10 and 11 for Ross or 1, 2 and 6 for the Antarctic Peninsula)?

For the IB experiments, combining using the current bounds is a bit tricky, because in many cases the different regions are assigned different bounds. This is one of the reasons why we use the UB experiments instead, to test the effect of partition choice. Changing the bounds when combining partitions would have its own effect on the final distribution, and this would make it more difficult to actually isolate the effect of partition choice. For example, with the current partition experiments shown here, we have already detected a drift caused by accumulation forcing, which in a sense complicated our analysis (as discussed at the end of the effects of partition choice section). However, with regards to the current IB sampling results, we would like to point out that the distributions shown do already include realizations where changes in proximal regions correlate with each other (which would be the imposed constraint if partitions were combined). They just also happen to include a number of samples where these partitions do not correlate, as well.

Therefore, our results do offer enough information to hypothesize about your interesting questions. With regards to IBAccumOnly, what we did learn from the partition experiments is that combining regions for accumulation sampling decreases the mean SLE contribution, since over a short (100 year) timescales mass storage dominates over ice dynamic response to accumulation change. With regards to the MeltOnly experiments, combining neighbors results in a smaller subset of neutral samples (when the regions may cancel each other out) and allows for a larger subset of endmembers (when the regions combine), so we expect an overall larger spread in uncertainty as partitions are combined. When combining low elevation and high elevation partitions within the same basin, we expect the expansion of right SLE distribution tails to be affected even more, as the dynamic response of the upper partition would be highly dependent on the grounding line retreat of the lower partition. We have added some sentences in the discussion to summarize these results and to point out the limitations of our strategy results.

This is definitely a very interesting and complex part of designing UQ studies for projections, and should be investigated further. We look forward answering these questions in the future, using an experimental setup to isolate these phenomena. Certainly, choosing an appropriate partition configuration is very important now, as discussed in the manuscript, and will become more important as this method matures. We have added to the uncertainty quantification methods section, to explain why we chose our particular configuration, in that sampling of all variables together requires a reconciled experimental design.

Minor comments

- page 1, line 13: Maybe omit 'instantaneous' here, since you do not test for ice-shelf collapse during the model run and your statement could be read to say 'it is not possible to achieve $> 1.2$ meters with a collapse during the run'.

Thank you for this suggestion. We have removed the word 'instantaneous'.

- page 1, line 14 and line 17, page 2, line 25,...: Using the term 'scenario' could be misleading. I did understand this first in the sense that you are making projections of Antarctica's future sea-level contribution, while you are, in fact, assessing uncertainties by using idealized, step-forcing experiments.

We have updated the manuscript to reflect this suggestion. Scenario is now used only to describe IPCC realizations or projections. We now refer to our IB and UB scenarios as 'experiments', and have changed the wording in other cases using the term 'scenario' throughout the manuscript.
- page 1, line 15: Maybe combine the sentences to make clear that this estimate is linked to the UB experiment?

In the abstract, these sentences have now been combined.

- page 2, line 6: Maybe add that 30cm was the upper bound in *Ritz et al.* (2015) and that > 1m was found for RCP8.5 by *DeConto and Pollard* (2016).

We have updated this sentence to reflect the fact that these two studies used different warming scenarios, and that the *Ritz et al.* (2015) upper bound was reported as 30 cm.

- page 3, line 18: What do you mean by 'coast' - the grounding line or the ice front? How large is the resolution around the grounding line (see also page 17, line 21). How is the resolution in areas where the grounding line retreats during the model run? Do you refine your mesh during grounding line retreat?

We have updated this sentence to specify that the 1 km is along the domain boundary and also along the shear margins. We do not have the capability to refine our mesh through time, so the resolution stays fixed as represented in Fig. S1. Please see our above response to Reviewer 1, regarding this issue. We have added a number of statements throughout the text to clarify that our interior mesh resolution is lower than 8 km, and that this is a caveat that should be considered by the reader.

- page 3, line 5: What model parameters do you use, e.g., related to Glen's flow law or the Budd-type friction law?

Details regarding the friction law used, and Glen's flow law exponent have been added to the text here.

- page 3, line 29: Why do you invert for ice viscosity in the floating areas instead of applying the same procedure as for the grounded areas (assuming thermal steady state)? Does this introduce jumps in ice viscosity across the grounding line?

We invert for ice viscosity in order to constrain the surface velocities of the ice shelves, after common practice for realistic forward runs using ISSM in Antarctica (e.g. *Seroussi et al.* (2017)). Since there is no basal friction on ice shelves and ISSM does not use an enhancement factor, this is how we constrain the ice shelf velocities. This step results in velocities that look as close as possible to observed. This inversion technique takes into consideration more than just temperature in determining ice shelf rigidity, and as discussed in the manuscript, it can also represent damage in our formulation (*Borstad et al.*, 2013). This field can indeed be discontinuous, and in some cases there are jumps in ice viscosity across the grounding line. With regards to this study, since we focus on grounding line retreat, we do not anticipate that our results would vary dramatically if we constrained ice shelf velocities using a different technique.

- page 4, line 21: Are the exchange parameters optimized per shelf or per region or continentwide?

The exchange parameters are optimized per shelf. We now specify this in the text.

- page 4, line 32: Please clarify the 'large melt multipliers in the ice sheet interior'.

We agree that this sentence does not make much sense. We have revised it to explain that our nearest neighbor schemes likely "over-estimate interior melt rates, leading to overly aggressive grounding line retreat, especially in the ice sheet interior".

- page 5, line 5: Are there regions where the grounding line advances? How do you specify basal friction in these areas?

There are some places where advance occurs, but as noted, only minor changes are observed. Friction is specified in all areas through inversion techniques (as done for the grounded parts of the ice sheet). For the basal friction inversion, these areas are assumed to be grounded and maintain an observed surface topography.

- page 5, line 19: This is a very interesting and important assessment. I wonder if (just for curiosity, if you did not do this, I do not ask you do to so): did you also compare to the original Bedmap2?

Yes, we did run with the original Bedmap2. Results were similar to the results presented here for the Bedmap2/MC: enough so that they were not really worth showing. If we can run in the future with much higher interior mesh resolution, we may be able to detect significant differences, which would indeed be very interesting!

- Section 2.4: Please give more details on the underlying assumptions, capabilities and limitations of the UQ method.

As discussed above (in response to the major comment) and in our responses below, we have expanded upon our UQ method in this methods section, the effect of partition choice results section, as well in the discussion paragraph that notes limitations associated with partition configuration and step function forcing. We hope that these additional sentences now offer satisfactory details about the ISSM-DAKOTA framework and the UQ strategy applied here (and associated assumptions).

- Section 2.4: In this section you relate to 'errors' in the model boundary conditions (e.g., lines 21, 24, 28). This might be confusing the reader since you quantify the effect of uncertainties in the variables rather than errors.

We have changed the reference to errors in this section to be 'uncertainties' instead.

- page 5, line 23: Reading your manuscript, I did understand that you use Latin-Hypercube sampling and not Monte-Carlo sampling (also stated in the abstract)?

We have removed references to Monte-Carlo style sampling in the manuscript.

- page 5, line 27: You state that you assume a uniform distribution of uncertainties to sample the tested variables (sub-shelf melt rates, surface accumulation, ice viscosity and basal friction). This sounds reasonable given that we do not know how

future greenhouse gas emissions evolve or which RCP scenario is most likely and how boundary conditions change in the future. Could you elaborate more on this in the manuscript: why did you chose this type of distribution and how does this affect your results, e.g., in comparison to using normal distributions?

Here, we have expanded our discussion about the uniform distribution. We have added some sentences describing our justification for our choice, and have discussed that we believe, because of the use of LHS, our results would not greatly differ if we chose a different forcing distribution (i.e. normal). Past experience with LHS suggests that for forward simulations, results of sampling with either distribution does not greatly change our results (*Schlegel et al.*, 2013). However, since our goal is to better quantify the probability of endmember occurrence (or the 95% confidence interval), we believe that uniform sampling is the most appropriate choice.

- page 5, line 31: Is there a specific reason that partitions should have equal areas? And how does it affect the IB-results that the GP do not have equal areas?

The partitions do not have to be equally weighted, but because our mesh is anisotropic, a random sampling by independent vertices would numerically weight every vertex unequally. Therefore, our practice is to sample by equal area to eliminate biases. This is discussed and tested by *Larour et al.* (2012b). Such a practice is most important when one wants to eliminate spatial correlations. However, as discussed in the next paragraph, we believe that for the purpose of future projections, it is likely important to represent some spatial correlation. At such a large scale, it certainly does affect the IB results that the GP is not equal area, since some basins will technically be 'weighted' more than others. However, we feel this is more realistic considering our restrictive knowledge about the future. We feel this is actually one of our more interesting outcomes from the study presented here, and it will certainly shape our methods

for continental-scale UQ in the future! In the text, we have rephrased these sentences and added a reference for clarity.

- page 6, line 3: Please explain.

In this section, we have expanded upon our explanation of how we choose the geographic partitions, including inference of climate forcing change for all the variables that informed our partitioning.

- page 6, line 11: Could you solve the problem of how to define reasonable partitions by specifying forcing scenarios for the variables (where possible, e.g. for the surface accumulation based on RCP scenario projections) and by using the UQ method to quantify uncertainties imposed on these?

Our partition choices were, to an extent, based on warming scenario results as noted in the IB bounds determination section. Yet, because we need to choose partitions that represent variability for all forcing variables being sampled, choosing appropriate partitions is not a straightforward task. This is especially the case considering that, informed by our partition analysis, we have decided to maintain the spatial correlation of ice flow within drainage basins for this study. The restriction of the same partitions for all variables is definitely a limitation in our method, and we have now noted this in the text. Due to this study, many questions have now surfaced about experimental design for UQ analysis, and how to answer them appropriately. In the future, answering these types of questions could constitute its own area of investigation. Because ISSM is open source, it has the potential to be a community effort, and will likely require much more analysis beyond what is presented here.

- page 7, line 24: Can you give an estimate to what amount of surface warming this extreme scenario corresponds?

We have added a statement that surface water requires a warming of about 30 degrees C in the interior and 5 degrees C along the margins of the ice sheet.

- page 9, line 5: How large do melt rates of FRIS and in the Amundsen sea get?

Please see Table 2, which notes the mean upper melt rates for every partition for both the UB and IB experiments.

- page 9, line 15-20: You could state here, that these bounds also encompass all RCP projections (if this is correct). This would underline that you are not only interested in the strongest emission scenario, but that you quantify uncertainties encompassing all scenarios.

We have added a sentence in this paragraph noting that our IB upper and lower accumulation bounds encompass the full spread of all projections.

- page 10, line 17: Is it valid to assume that variables can be sampled independently from each other and that partitions can be sampled independently? Please add a discussion.

This paragraph has been removed from the manuscript, as suggested by Reviewer 1, and we agree that the wording repeats (and complicates) the explanation of our methods. As described in the methods, each variable is sampled individually, as well as synchronously, so 'independently' is indeed a misleading word to use in this context.

- page 13, line 7: mass 'gain' instead of 'loss'.

Thank you. We have updated this to read 'gain'.

- page 13, line 31: The bi-modality is very interesting - do you have further hints on how this arises? Is it linked to instability within a specific basin or ice stream?

Yes, this behavior is linked to instability within a subset of regions. To truly diagnose and illustrate this phenomenon, we need to conduct a spatial analysis which is out of the scope of the investigation presented here. We are working on this analysis for a separate publication, as we also find it to be a very interesting result!

- page 14, line 26: How large are melt rates around the grounding line in your control run?

We have added a table in the Supplementary material. This new table gives information for mean melt rates and standard deviation within each partition as well as the rates at the grounding line.

- page 14, line 29: How does the mean SLE from the Amundsen Sea Region for the IB experiment compare to observed rates of sea-level rise (e.g. *Shepherd et al.* (2018))?

Thank you for this suggestion. The mean SLE contribution is 6 mm over the 100 year simulation. We have added a sentence to note that this value is equivalent to the acceleration in mass loss already observed during the last decade in WAIS.

- page 14, line 35: The intrusion of relatively warm water into the Filchner-Ronne ice shelf cavity has been found possible within the second half of the 21st century for the RCP8.5 scenario (*Hellmer et al.*, 2012). This would be much later than in the step forcing experiments applied here - how is this reflected within your uncertainty quantification method?

This is a good point. Our method does not yet test the timing of an extreme change in climate forcing. Instead, our endmembers represent the most extreme realizations possible within 100 years (simultaneous and instantaneous change), and they are maintained for a century. We have added a note in the methods section to emphasize that our forcing is a step function. In addition, we have rephrased this sentence to point out that our forcing is based on ocean sensitivity experiments that reveal sensitivity to Ronne Ice Shelf melt rates, similar to that suggested by *Hellmer et al.* (2012). Finally, in the discussion, where we discuss the step function potential overestimation of SLE contribution, we have added a statement about future use of this framework to quantify uncertainty in more complex, forward simulations forced with warming scenarios.

- page 15, line 3: Could you add a figure with an example for grounding line migration?

We have expanded the Ronne region in the supplementary figure, and grounding line migration that is included. The expanded region should now be sufficiently large enough to show extreme grounding line changes in Institute and Möller Ice Streams.

- page 15, line 16: Maybe reformulate this to 'subsequent reduction in buttressing'. The current formulation might cause misunderstanding, since you account for the reduction of buttressing through changes in sub-shelf melting.

The text has been update as suggested.

- page 17, line 8: Please explain what you mean with 'well-behaved' and 'normally'? Does this relate to the next sentence about the bimodal structure?

We have updated this sentence to say that the SLE curves are normally distributed, even though we are forcing with a uniform distribution.

- page 17, line 15: E.g., shorten to '... bedrock geometry and ice-shelf buttressing, which determine grounding line stability.'

Thank you. This sentence as been changed as suggested.

- page 18, line 10: Why does your reasoning presented here for the UB experiments (lower resolution provides less bed geometry details and this yields more SLE) not apply to the IB experiments?

We have revised this paragraph to specifically describe the difference between the low resolution IB and UB results, including the fact that PIG and Wilkes Land grounding line are highly affected by the resolution of the bedrock geometry. Therefore, these areas are more stable in our lower mesh resolution, less extreme IB simulations.

- page 19, line 14: Another process missing might be isostatic rebound, as recent evidence suggests *Barletta et al.* (2018).

We have added a sentence noting that we are not considering solid earth feedbacks here.

- Appendix A, page 21, line 31: This setup was created as an idealized version of the cavity underneath Pine Island Glacier ice shelf. Did you test also for cavity sizes and shapes comparable to FRIS?

Because initial sensitivity experiments suggested that temperature effects dominated over both cavity size and shape, we did not test for a FRIS style cavity size and shape. This decision was made largely because, due to computational restrictions, we had to prioritize our ocean simulations. However, as a compromise we did expand the tested

temperature range (-2C to +3C), which based on our initial sensitivity experiments, we deemed adequate for realizing the sensitivity of both cold and warm water ice shelves.

- Appendix A, page 22, line 5: 'average ice shelf melt rate'?

We have added the word 'average', as suggested.

- Appendix A, page 22, line 23: Please clarify 'that does not artificaly restrict ice flow.

This sentence now specifies that the melt rates are restricted to values that do not "result in ice mass removal that exceeds the local mass balance between the divergence of ice flow and surface mass balance".

- Table 2: Please specify 'mean max melting rate'.

The caption for this table now specifies that, "The mean upper bound melt rate is equal to the maximum melting rate multiplier times the control ice shelf melting rate values, area averaged over the initialization ice shelf area within each margin partition."

- Figure S1: The upper panel looks squeezed in y-direction. The line colors to indicate the sectors in the lower panels are hard to see.

We have updated this figure such that the axes are now the same as our other Antarctica figures. In addition, we have thickened the lines for the regional boxes, such they should be much easier to see now.

- Figure S5: It is hard to see the grounding line positions, especially the initial position.

In these figures, we have updated the color scheme, thickened the grounding position lines, and have changed the color of the initial position to yellow. The information in these plots should now be much easier for the reader to see.

Technical issues

- page 4, line 1: remove 'are' before 'described'.

'are' has been removed from the manuscript.

- page 10, line 7: 'actual uncertainty'

'actually' has been changed to 'actual', as noted
- page 17, line 10: There is a verb missing in '.., as evidence by..'.

This sentence has been updated to read "Specifically, forcing with extreme ice shelf melt rate bounds results in a total AIS SLE contribution distribution that is bimodal, which illustrates the dynamic complexity of grounding line response."

- page 20, line 12: This sentence is doubled.

We have removed the second part of this sentence that is repetitive. It is now updated to read "In contrast, our more plausible IB scenario results show relatively minor contribution from the Amundsen Sea."

- page 21, line 29: 'setup'

This change has been made as suggested.

**References**

Barletta, V. R., et al., Observed rapid bedrock uplift in Amundsen Sea Embayment promotes ice-sheet stability, *Science*, *360*, 1335–1339, 2018.

Borstad, C. P., E. Rignot, J. Mouginot, and M. P. Schodlok, Creep deformation and buttressing capacity of damaged ice shelves: theory and application to Larsen C ice shelf, *Cryosphere*, *7*, 1931–1947, doi:10.5194/tc-7-1931-2013, 2013.

DeConto, R., and D. Pollard, Contribution of Antarctica to past and future sea-level rise, *Nature*, *531*, 591–597, doi:10.1038/nature17145, 2016.

Hellmer, H. H., F. Kauker, R. Timmermann, J. Determann, and J. Rae, Twenty-first-century warming of a large Antarctic ice-shelf cavity by a redirected coastal current, *Nature*, *485*(7397), 225–228, doi:10.1038/nature11064, 2012.

Larour, E., and N. Schlegel, On ISSM and leveraging the Cloud towards faster quantification of the uncertainty in ice-sheet mass balance projections, *Comp. Geosci.*, *96*, 193 – 201, doi:http://dx.doi.org/10.1016/j.cageo.2016.08.007, 2016.

Larour, E., M. Morlighem, H. Seroussi, J. Schiermeier, and E. Rignot, Ice flow sensitivity to geothermal heat flux of Pine Island Glacier, Antarctica, *J. Geophys. Res. - Earth Surface*, *117*(F04023), 1–12, doi:10.1029/2012JF002371, 2012a.

Larour, E., J. Schiermeier, E. Rignot, H. Seroussi, M. Morlighem, and J. Paden, Sensitivity Analysis of Pine Island Glacier ice flow using ISSM and DAKOTA, *J. Geophys. Res.*, *117, F02009*, 1–16, doi:10.1029/2011JF002146, 2012b.

Rignot, E., S. Jacobs, J. Mouginot, and B. Scheuchl, Ice shelf melting around Antarctica, *Science*, *341*(6143), 266–270, doi:10.1126/science.1235798, 2013.

Ritz, C., T. Edwards, G. Durand, A. Payne, P. V., and R. Hindmarsh, Potential sea-level rise from Antarctic ice-sheet instability constrained by observations, *Nature*, *528*(7580), doi:10.1038/nature16147, 2015.

Schlegel, N.-J., E. Larour, H. Seroussi, M. Morlighem, and J. E. Box, Decadal-scale sensitivity of Northeast Greenland ice flow to errors in surface mass balance using ISSM, *J. Geophys. Res. - Earth Surface*, *118*, 1–14, doi:10.1002/jgrf.20062, 2013.

Schlegel, N.-J., E. Larour, H. Seroussi, M. Morlighem, and J. E. Box, Ice discharge uncertainties in Northeast Greenland from boundary conditions and climate forcing of an ice flow model, *J. Geophys. Res. - Earth Surface*, *120*(1), 29–54, doi:10.1002/2014JF003359, 2015.

Schodlok, M., D. Menemenlis, and E. Rignot, Ice shelf basal melt rates around antarctica from
simulations and observations, *J. Geophys. Res.*, *121*, 1085–1109, doi:10.1002/2015JC011117, 2016.

Seroussi, H., Y. Nakayama, E. Larour, D. Menemenlis, M. Morlighem, E. Rignot, and A. Khazendar, Continued retreat of Thwaites Glacier, West Antarctica, controlled by bed topography and ocean circulation, *Geophys. Res. Lett.*, *44*, doi:10.1002/2017GL072910, 2017GL072910, 2017.

Shepherd, A., et al., Mass balance of the Antarctic ice sheet from 1992 to 2017, *Nature*, *558*(7709), 219+, doi:10.1038/s41586-018-0179-y, 2018.

---

## Referee Report (RR1)

**Comment on the revised manuscript 'Exploration of Antarctic Ice Sheet 100-year contribution to sea level rise and associated model uncertainties using the ISSM framework'**

by Schlegel et al.

Thank you for your thorough revision of the manuscript and addressing most comments from both reviews! However, I think that the major comment raised about the UQ methodology has not been addressed fully:

I agree with you, that climatological uncertainties are more complex than error maps because of the strong spatial correlation and the large spread and I appreciate that you added discussion on this to the manuscript. However, I think an important point is missing. The spatial correlation of the variables sampled makes the use of the UQ method as presented in the manuscript questionable, since the UQ method is based on the assumptions that the variables can be sampled independently (here 'variable' relates to one of the four physical variables, i.e., basal melting, surface accumulation, ice viscosity, basal friction, sampled in one of the partitions). This assumption that the variables are independent of each other is certainly not true for climatological uncertainties in two ways: 1) the physical variables are spatially correlated, i.e., if atmospheric warming causes surface accumulation increases in the Bellingshausen Sea, surface accumulation will likely also increase in the Amundsen Sea region; and 2) the physical variables are not independent from each other, i.e., their changes will be some kind of monotonic function of mean global temperature changes. Hence, applying the UQ method as presented in the manuscript is methodologically questionable. I think that it would be better to re-design the experiments (see below) or at least to test and discuss the effects of this (also for the IB experiments).

Testing: I understand that changing the partitions is difficult since IB are chosen on a partitionlevel (see response on page C12). However, testing the combination of the low and high surface elevation partitions for the IBs should be possible in the IBMeltOnly experiment: in this case, only melt rates are sampled and these bounds are assumed to be similar in the coastal and the adjacent upstream partitions, as explained on page 24, line 14-15. If I understood this correctly, this would provide a simple case to test and discuss the effect of violating the basic assumption of the UQ method about independent variables. For other experiments, the bounds of to-bejoined partitions could be combined so that the variables are sampled in the same manner within both partitions but still within the original bounds, e.g., the physical variable is perturbed in both partitions by the value at x% of the interval between the lower and upper bound of the corresponding partition.

Testing this is especially important, since, as you point out in your response (page C12): 'We would like to point out that the distributions shown do already include realizations where changes in proximal regions correlate with each other. They just also happen to include a number of samples where these partitions do not correlate, as well.' I agree with you, that these realizations are already included in the results, however, by including these 'neutral outcomes' (where partitions are uncorrelated), your resulting probability density functions could be strongly biased.

Experimental redesign: A better way to apply the UQ method could be on the basis of 'error maps'. The experiments are designed as continent-wide forcings with climate warming scenarios, e.g. based on the different RCP scenarios, and by using the UQ methodology to analyse the effect of uncertainties imposed on these forcings. In this case, I think that it would be more plausible to assume that the uncertainties can be sampled like 'error maps' and are independent from each other.

Further issues:

- p1, line 6: Large uncertainties come also from the future GHG emission pathway taken by societies.
- p1, line 9: 'model simulations' rather than projections.
- page 6, line 7 and page 15, line 15-17: This is surprising, since the sampling method used (i.e. LHS or another method) should theoretically not influence the results, as it is only a technique to reduce the number of samples required to obtain a reliable uncertainty distribution. You state that you tested a larger number of samples did you find convergence of the resulting PDFs for increasing the sample size? Or could this be related to the fact that you use a rather small number of samples for the large number of variables tested (e.g. 108 variables in the case of 4 physical variables and 27 partitions)?
- At the same time, it is interesting that you find the distribution of the variables sampled to have no affect on your results. Do you have an idea how this could be explained?
- Please add to the discussion the point raised above: "The UQ methodology is based on the assumption that the physical variables can be sampled independently from each other and in-between the partitions. This is in contrast to the fact that climatological

uncertainties, as tested here, are spatially correlated. In consequence, the distributions of SLE contribution obtained here could be strongly biased and, for example, significantly underestimate the spread in uncertainties."

• page 10, line 30: This seems to be a misunderstanding, since I did wonder if also the other RCP scenarios (2.6, 4.5) are included?

---

## Author Response (AR2)

Thank you for continued support and interest in our manuscript. Discussions concerning our methods are quite important, and below, we try to address all additional questions and concerns. Again, for comments that required updates to text and/or figures, we enclose a draft with corresponding changes in red.

**1    Editor**

Comments to the Author:

Dear Nicole-Jeanne,

The revised version of your manuscript has been re-reviewed by one of the reviewer from the first review step. If the reviewer acknowledges that most of the comments have been addressed, there is still a major comment that should be addressed. I have also read this new version and found some minor typos that should be corrected in the new version. Based on this new version of your paper, I will then take a final decision.

Regards, Olivier Gagliardini

Minor points:

- page 3, line 11: it should be clearly mentioned that a perfect connexion with the ocean is assumed for the basal hydrology

We have added a statement to clarify that we area "assuming a perfect connection with the ocean and no hydrological flow".

- page 6, line 19: to be regions of => to be continuous regions of ? (even if it is clear from the supplementary figures)

The word continuous has been inserted where suggested.

- page 11, line 19: this point is also addressed in Pattyn (2017)

A reference to *Pattyn* [2017] has been added here.

- caption Table 2: make a reference to Fig. 1 for the definition of the 27 GP.

A reference to Fig. 1 has been added to this caption.

- page 15, line 14: the mean global mean sea level (mean mean?)

This sentence has been updated to read 'mean SLE' instead, for clarity.

- page 17, line 34: mass loss decrease due the Bedmap1 (due to?)

The word 'to' has been inserted as suggested.

**2 Anonymous Referee 2**

Thank you for your thorough revision of the manuscript and addressing most comments from both reviews! However, I think that the major comment raised about the UQ methodology has not been addressed fully:
I agree with you, that climatological uncertainties are more complex than error maps because of the strong spatial correlation and the large spread and I appreciate that you added discussion on this to the manuscript. However, I think an important point is missing. The spatial correlation of the variables sampled makes the use of the UQ method as presented in the manuscript questionable, since the UQ method is based on the assumptions that the variables can be sampled independently (here 'variable' relates to one of the four physical variables, i.e., basal melting, surface accumulation, ice viscosity, basal friction, sampled in one of the partitions). This assumption that the variables are independent of each other is certainly not true for climatological uncertainties in two ways: 1) the physical variables are spatially correlated, i.e., if atmospheric warming causes surface accumulation increases in the Bellingshausen Sea, surface accumulation will likely also increase in the Amundsen Sea region; and 2) the physical variables are not independent from each other, i.e., their changes will be some kind of monotonic function of mean global temperature changes. Hence, applying the UQ method as presented in the manuscript is methodologically questionable. I think that it would be better to re-design the experiments (see below) or at least to test and discuss the effects of this (also for the IB experiments).

Yes, we can definitely try test the effects of making specific partition choices using the IB sampling. In order to do so, 1) we have taken your suggestion below and removed the elevation separation for the 27 partitions (resulting in 18 partitions) for the IBMeltOnly experiment, and 2) we have designed a new set of experiments which force a spatial correlation as well as a variable correlation, in order to aid in the quantification of these effects.

In the newly designed set of experiments (IBCorr_27GP and UBCorr_27GP), for all 27 partitions, accumulation and melt are only allowed to vary within the upper 25% of their uncertainty bounds while viscosity and friction are only allowed to vary within the lower 25% of their uncertainty bounds. This experiment is repeated for the 25-50%, 50-75%, and 75-100% ranges. Note that

these relationships are chosen in the absence of any well-established physical representation in the literature of how these variables might actually correlate with warming. After the runs are completed, results are combined to create a new distribution which forces spatial and variable correlation. These results are compared to those of our original results.

We have designed these experiments with the goal of quantifying the worst case effect of variable and spatial correlation. However, it is important to note that we also had to consider the fact that each of these runs are very expensive, as previously noted in the manuscript (in dollar amounts up to thousands of dollars each). Because of this obstacle, we cannot complete a large suite of additional experiments at this time. New experiments that test model uncertainty in a different way may be a possibility in the future, and we certainly appreciate helpful suggestions for different ways to approach UQ strategy using our framework. In light of our resource limitation currently, we have attempted to quantify correlations in the most efficient way possible. We hope that the results presented below are sufficient, and thank you for your understanding in this matter.

Testing: I understand that changing the partitions is difficult since IB are chosen on a partitionlevel (see response on page C12). However, testing the combination of the low and high surface elevation partitions for the IBs should be possible in the IBMeltOnly experiment: in this case, only melt rates are sampled and these bounds are assumed to be similar in the coastal and the adjacent upstream partitions, as explained on page 24, line 14-15. If I understood this correctly, this would provide a simple case to test and discuss the effect of violating the basic assumption of the UQ method about independent variables. For other experiments, the bounds of to-bejoined partitions could be combined so that the variables are sampled in the same manner within both partitions but still within the original bounds, e.g., the physical variable is perturbed in both partitions by the value at x% of the interval between the lower and upper bound of the corresponding partition.

Testing this is especially important, since, as you point out in your response (page C12): 'We would like to point out that the distributions shown do already include realizations where changes in proximal regions correlate with each other. They just also happen to include a number of samples where these partitions do not correlate, as well.' I agree with you, that these realizations are already included in the results, however, by including these 'neutral outcomes' (where partitions are uncorrelated), your resulting probability density functions could be strongly biased.

As discussed above, we have tested the IBMeltOnly experiment using a smaller number of partitions, such that drainage basins are not divided by elevation. Considering for the IB case that the Melt is responsible for the largest uncertainty, this experiment should test a large percentage of bias that might be

[Figure]

Figure 1: PDF of Antarctica SLE (m) contribution after 100 years, produced by sampling ice shelf melt rates with IB, using 27 GP (red) compared with the use of 18 Geographical Partitions (black) where partitions within the same drainage basin are combined and not divided by elevation.

introduced by our decision to separate the basins by elevation. We show a comparison between the original IBMeltOnly with 27 GP and the new 'combined' 18 Geographical Partition run in Fig. 1. Here, we illustrate that the results (mean and uncertainty) vary by less than 1% between the 2 runs. Likely, this is due the fact that the grounding line, even in the most extreme warming case, does not retreat beyond the present day 2000 m surface elevation contour (e.g. Supp. Fig. S5 in the manuscript) and that the largest contributing region to uncertainty (i.e. Ronne) is not divided by elevation for the original 27 GP. We have added a statement to the text to note this as a justification for the elevation separation of our partitions.

For the second set of experiments, forcing variable and spatial correlations, we have conducted the experiments for both UB_27GP and IB_27GP (i.e. IB-Corr_27GP and UBCorr_27GP, Fig. 2). The results from the IBCorr_27GP experiment suggest that forcing such correlation does affect the uncertainty spread of the PDF curve, reducing it by 0.065 m. This is mostly because of the interplay between accumulation and iceshelf basal melt rates (since increases in accumulation and now forced to correlate with increased melt rates, and changes ice friction and viscosity are minor compared to melt rates). For the mean, we find a shift to the left. This results is not surprising, as it follows the phenomenon explored in the manuscript (i.e. a shift to the left observed when number of partitions are reduced). In this case, the storage of accumulation is due to spatial correlation between the 27 partitions as well as the fact that in the interior, accumulation affects mass balance while ice shelf basal melt rates do not. While slightly different from the shift due to a spatial correlation forced when running with a smaller number of partitions (as discussed in the manuscript), both results are sourced in the correlated storage of mass on the continental ice sheet.

The UBCorr_27GP results are much different. As suggested in the manuscript

[Figure]

Figure 2: PDF of Antarctica SLE (m) contribution after 100 years, produced by sampling ice shelf melt rates with IB (red, above) and UB (red, below), using 27 GP compared with a sampling experiment designed to force variable and spatial correlations (black): IBCorr_27GP and UBCorr_27GP as described in our response above.

by our UB_1 results, forcing spatial correlation widens our uncertainty. Here, the forced correlation between accumulation and ice shelf basal melt rates does also mitigate the spread of the PDF curve (as compared to UB_1); however, in the more extreme realizations (high melt/high accumulation/low viscosity and friction or low melt/low accumulation/high viscosity and friction), the dynamic grounding line response driven by ice shelf basal melt rates (particularly in the Amundsen Sea), as well as lower friction and lower ice viscosity, are not completely mitigated by accumulation. For the mean, similar to the IBCorr_27GP experiments, the UBCorr_27GP also experiences a shift to the left.

To illustrate the effect of only variable correlation, we have also run the same experiments for UB_1 (UBCorr_1, Fig. 3). Results for the one partition experiment do not force additional spatial correlation, only variable correlation, since spatial correlation is already built into the one partition design. Here, we observe a very minor narrowing of the uncertainty bounds and a shift to the left. These results suggest that for the UBCorr_27GP results (Fig. 2, bottom), the forced spatial correlation plays a significant role in widening the uncertainty

[Figure]

Figure 3: As for Fig. 2, but for the UB_1 experiment only: UBCorr_1.

bounds (since widening does not occur in the UBCorr_1 case). In contrast, for IBCorr_27GP, the informed bounds for the combined variables do not expose a dynamically sensitive response on the continental scale. In this case (Fig. 2, top), results suggest that accumulation's mitigation of SLE contribution from grounding line retreat is the more dominant process.

A summary of these results has been added to the manuscript discussion.

Experimental redesign: A better way to apply the UQ method could be on the basis of 'error maps'. The experiments are designed as continent-wide forcings with climate warming scenarios, e.g. based on the different RCP scenarios, and by using the UQ methodology to analyse the effect of uncertainties imposed on these forcings. In this case, I think that it would be more plausible to assume that the uncertainties can be sampled like 'error maps' and are independent from each other.

As noted above, a current redesign is not possible. Such a study would constitute an entirely different study, with a goal of quantifying the effects of scenario variability. While each sampling experiment independently costs about 65k CPU hours (which are resources unfortunately not available to us at this time), in the future, we would like to expand our study to sample models with increased complexities (i.e. transients). We believe the the first steps - developing a better understand of how different forcing affect uncertainty using a simple simulation - are addressed in the current manuscript. As a follow-up, it would definitely make sense to conduct a study forced with various RCP scenarios. We will certainty consider your suggestions in the future. As discussed above, in lieu of redesign, we have tried to design new experiments to study/quantify the effects of correlations, and we hope that they sufficiently answer the questions posed regarding our methods.

Further issues:
- p1, line 6: Large uncertainties come also from the future GHG emission pathway taken by societies.

We have added a statement in the abstract that reflects that: the choice of emission scenario also contributes to uncertainty in ice sheet model projections.

- p1, line 9: 'model simulations' rather than projections.

The abstract has been updated as suggested.

- page 6, line 7 and page 15, line 15-17: This is surprising, since the sampling method used (i.e. LHS or another method) should theoretically not influence the results, as it is only a technique to reduce the number of samples required to obtain a reliable uncertainty distribution. You state that you tested a larger number of samples - did you find convergence of the resulting PDFs for increasing the sample size? Or could this be related to the fact that you use a rather small number of samples for the large number of variables tested (e.g. 108 variables in the case of 4 physical variables and 27 partitions)?

We agree that the referenced statement is misleading, as we indeed do not expect the use of LHS to affect our results. Our statement was only meant to point out that LHS, uniform sampling, should emphasize the extremes (tails) - more than a normal pure random sampling would, allowing the use of less samples to robustly represent the resulting PDF. We did not intend to suggest that the method would influence our results beyond requiring less samples. In fact, as you suggest, we strive for a method that is both efficient and accurately represents the PDF that we would have been acquired if we had infinite resources. The sentence has been rephrased, so that it is more clear that we want to make sure extreme cases are realized while using less number of samples.

During our experiments varying the number of samples, the PDF curves do indeed converge with increasing number of samples, as expected and previously illustrated by *Larour et al.* [2012]. We find that with 800 samples, we can capture the mean and spread of the PDF acquired with up to 3200 samples to within 0.6% (See Fig. 4). We have updated the text to reflect this point and make it more clear to the reader.

- At the same time, it is interesting that you find the distribution of the variables sampled to have no affect on your results. Do you have an idea how this could be explained?

This is a good and important point you bring up here. We state that in past studies, the distribution of sampled variables has had little effect on our results,

[Figure]

Figure 4: PDF of Antarctica SLE (m) contribution after 100 years, sampled with IB (above) and UB (below) with various number of sample simulations.

and that normal and uniform distributions have yielded similar conclusions. This statement is based on studies where we use a large number of partitions

(e.g. *Schlegel et al.* [2013]), and therefore in general, we find that with an increase in partitions, the mass forcing PDFs in particular become more normal and more narrow (as is dictated by the Law of Large Numbers and the Central Limit Theorem). This occurs regardless of the actual shape of our original distribution (as long as the bounds are consistent). Therefore, with a generally large number of partitions, the distribution curves of the model mass forcing itself (total SMB and total basal mass balance) are actually similar whether the partition-based uncertainty distribution is uniform or normal.

Since accumulation and melt rate are the largest contributors to mass balance and sea level contribution, theoretically in the absence of dynamic ice response, the larger the number of partitions and the larger number of samples, the more similar our uniform and normal sample total SLE contribution results should be. Based on the experiments conducted for this study, we find that this is indeed the case in practice. We have added some sentences to clarify this point in the partition results section. The text now emphasizes that the mass forcing is partly responsible for the narrowing of the PDF curves with increasing partitions. In addition, as discussed in previous versions of the manuscript, ice dynamics (e.g. proximal correlation) also partially drives the width of the total SLE response.

We do acknowledge that for this study, we use 27 partitions, which is certainly not many compared to the hundreds of partitions used in our past studies. This complicates our expectations, since with this smaller amount of partitions of a uniform distribution, there is certainty no guarantee that our total ice sheet mass forcing PDF curves (e.g. total SMB) will be normal, especially since 27 partitions is slightly less than the estimated minimum required for the Central Limit Theorem to strongly apply ($>=30$). We agree that this is an important caveat, and therefore we have investigated the shape of the total SMB forcing PDF curves for our 27 GP experiments. We find that these PDFs are indeed normally distributed, with very little skew. This suggests that a normal and uniform sampling scheme with the same spread should result in a similar total mass forcing on the entire ice sheet. Because of this, we do expect that the choice of uniform sampling should not greatly affect our total SLE contribution results (as stated in the manuscript), and that the use of a normal distribution would yield similar (though admittedly not identical) results.

- Please add to the discussion the point raised above: 'The UQ methodology is based on the assumption that the physical variables can be sampled independently from each other and in-between the partitions. This is in contrast to the fact that climatological uncertainties, as tested here, are spatially correlated. In consequence, the distributions of SLE contribution obtained here could be strongly biased and, for example, significantly underestimate the spread in uncertainties.'

Text has been added to the discussion, to point out that our results do not consider spatial or variable correlations. We note that in reality, there are

correlations between the variables we sample, and that our results may have bias because these relationships are ignored. Above, we present experiments that attempt to test the effects of spatial and variable correlation, and we have also included new statements in the discussion to quantify this caveat.

- page 10, line 30: This seems to be a misunderstanding, since I did wonder if also the other RCP scenarios (2.6, 4.5) are included?

Yes, this was a misunderstanding on our part. As stated in line 25, only the RCP8.5 scenarios, which we consider to be inclusive extreme scenarios, are used to derive endmember bounds for our sampling. The other scenarios are not considered. The sentence added during the last revision has been removed.

**References**

[revised manuscript text omitted]

**Supplement Figures**

**Table S1.** Mean melt rate estimates derived from satellite (2003-2008, Rignot et al. (2013)) and ocean model simulations with standard deviation (2004-2008) for each GP. We also include mean melt rate estimates at the grounding line, averaged over the control run period (2004-2013) for each GP. The last two columns list the mean cavity daily maximum temperature (cavity Tmax) and the mean daily maximum boundary temperature (Tmax) for 2004-2008. Cavity Tmax is defined as the mean maximum daily temperature within all ice shelf cavities of a particular GP. Tmax is defined as the mean maximum daily temperature below the thermocline and off of the continental shelf, and consists of Warm Deep Water in the Weddell Sea and Circumpolar Deep Water elsewhere. In derivation of our IB maximum melt multipliers, we assume that the difference between Tmax and cavity Tmax is representative of the maximum heat that could enter the cavity and potentially contribute to extreme melt rates (see Appendix Sect. A).

| Geographic partition number | Rignot et al., 2013 rate (m y$^{-1}$) | Mean Melt rate (m y$^{-1}$) | Mean Grounding Line Melt rate (m y$^{-1}$) | Mean Cavity Temperature Maximum (°C) | Mean Temperature Maximum (°C) |
|---|---|---|---|---|---|
| 1 | 1.1 | $0.94 \pm 0.61$ | $0.38 \pm 0.52$ | $-1.5$ | 0.68 |
| 2 | 2.78 | $2.77 \pm 0.36$ | $1.59 \pm 1.78$ | $+0.3$ | 2.32 |
| 3 | 2.6 | $3.28 \pm 1.78$ | $2.51 \pm 2.37$ | $-0.3$ | 2.27 |
| 4 | 11.3 | $11.83 \pm 2.35$ | $7.76 \pm 7.04$ | $+0.2$ | 2.27 |
| 5 | 2.15 | $3.48 \pm 0.24$ | $2.68 \pm 1.83$ | $+0.1$ | 2.12 |
| 6 | 0.85 | $0.86 \pm 0.14$ | $0.18 \pm 0.28$ | $-1.8$ | 0.76 |
| 7 | 0.3 | $0.37 \pm 0.06$ | $0.76 \pm 2.24$ | $-1.7$ | 0.67 |
| 8 | 0.4 | $0.4 \pm 0.03$ | $0.50 \pm 0.47$ | $-1.5$ | 0.64 |
| 9 | 0.27 | $0.38 \pm 0.16$ | $0.26 \pm 0.55$ | $-1.6$ | 0.47 |
| 10/11 | 1.74 | $1.56 \pm 0.56$ | $0.38 \pm 1.65$ | $-1.3$ | 1.38 |
| 12 | 2.18 | $1.95 \pm 0.65$ | $0.57 \pm 8.89$ | $-1.9$ | 0.97 |
| 13 | 0.56 | $0.63 \pm 0.30$ | $0.81 \pm 1.32$ | $-1.6$ | 0.92 |
| 19 | 2.35 | $2.51 \pm 0.64$ | $1.34 \pm 1.73$ | $-1.2$ | 1.19 |
| 24 | 6.83 | $0.89 \pm 0.74$ | $0.33 \pm 0.82$ | $-1.7$ | 1.34 |
| 25 | 0.6 | $0.64 \pm 0.16$ | $1.80 \pm 3.26$ | $-1.8$ | 1.04 |
| 26 | 1.95 | $1.56 \pm 1.61$ | $0.86 \pm 5.34$ | $-1.9$ | 2.00 |
| 27 | 4.1 | $3.84 \pm 1.05$ | $3.10 \pm 2.96$ | $-1.1$ | 2.17 |

**Model Mesh**

[Figure]

**Figure S1.** The ISSM Antarctica mesh for (a) the entire ice sheet, (b) Amundsen Sea sector (outlined in blue in (a)), and (c) Ronne Ice Shelf region (outlined in red in (a)). The green box shows the Wilkes Land (including Moscow University Ice Shelf and Totten Glacier) region (used in Fig. S5e,f).

[Figure]

**Figure S2.** Mean SLE contribution (m) from single 100-year simulations, under various extreme forcing, including uniform multiplication of ice shelf basal melt by 10 and 100; collapse of all ice shelves; uniform reduction of basal friction by 50% and 99.99%; uniform reduction of ice viscosity by 50% and 99.99%; and a combination of uniform extreme forcing (ice shelf basal melt multiplied by 10, basal friction decreased by 60%, ice viscosity decreased by 40%, and accumulation decreased by 50%). The combination results shown are equivalent to the max endmember of a UB 1-partition sampling experiment. (a) Comparison of results for these sensitivity experiments run for two different stress balance equations: L1L2 and SSA. (b) Percent difference between SSA and L1L2 simulations, corrected for the bias in their respective control runs. Note that the difference between all runs after 100 years is less than 5%.

[Figure]

**Figure S3.** Various partition configurations corresponding to the sampling experiments featured in Fig. 2, plotted over initial modeled surface ice velocities (m y$^{-1}$).

[Figure]

**Figure S4.** Comparison of SLE (m) PDFs for simulations performed with high (solid lines) and low (LowRes, dotted lines) mesh resolution. Ensemble runs of 800 simulations run with GP partitioning include UB combined variable (black) experiments, IB combined variable (red) experiments, and the UB melt only (dark blue) experiments.

[Figure]

**Control**

**Bedmap**

| | m |
|---|---|
| | 2000 |
| | 1000 |
| | 0 |
| | −1000 |
| | −2000 |

Initial Grounding Line
Uniform Bounds Grounding Line at 100 years
Informed Bounds Grounding Line at 100 years

**Figure S5.** Regional bed topography used in ISSM for simulations performed with the Bedmap2/MC (left panels) and Bedmap1 bedrock topographies (right panels). Regions included are: (a,b) Amundsen Sea, particularly Pine Island and Thwaites Glaciers (blue inset in Fig. S1), (c,d) Ronne Ice Shelf and ice streams (red inset in Fig. S1) and (e,f) Wilkes Land, including Moscow University Ice Shelf and Totten Glacier (green inset in Fig. S1). Yellow dashed lines are the initial grounding line positions. White solid lines and black solid lines are respectively the grounding line positions for the single extreme forcing highlighted in Fig. S6 for the UB and IB combined variable runs.

[Figure]

[Figure]

**Figure S6.** SLE contribution (m, with respect to a control run) from single AIS runs (200-year simulations) forced with extreme warming from the UB and IB experiments. Top: extreme endmember from the UB experiment (combination of ice shelf basal melt multiplied by 10, basal friction decreased by 60%, ice viscosity decreased by 40%, and accumulation decreased by 50%). Bottom: extreme endmember from the IB experiment (combination of parameter values set at informed extreme bounds, determined regionally). These two runs represent the regional SLE contribution far right endmember (maximum possible contribution) of the UB 1-partition (UB_1) and IB geographic partition (IB_27GP) sampling experiments. (See Table 1 and Table 2 respectively for details about simulation bounds.)